# Leveraging ancient DNA to uncover signals of natural selection in Europe lost due to admixture or drift

Devansh Pandey ®[1,6], Mariana Harris[2,6], Nandita R. Garud ®[3,4,7] ✉ & Vagheesh M. Narasimhan ®[1,5,7] ✉

Large ancient DNA (aDNA) studies offer the chance to examine genomic changes over time, providing direct insights into human evolution. While recent studies have used time-stratified aDNA for selection scans, most focus on single-locus methods. We conducted a multi-locus genotype scan on 708 samples spanning 7000 years of European history. We show that the G12 statistic, originally designed for unphased diploid data, can effectively detect selection in aDNA processed to create 'pseudo-haplotypes'. In simulations and at known positive control loci (e.g., lactase persistence), G12 outperforms the allele frequency-based selection statistic, SweepFinder2, previously used on aDNA. Applying our approach, we identified 14 candidate regions of selection across four time periods, with half the signals detectable only in the earliest period. Our findings suggest that selective events in European prehistory, including from the onset of animal domestication, have been obscured by neutral processes like genetic drift and demographic shifts such as admixture.

With the emergence of large sample-size sequencing data, numerous population genetic studies have attempted to identify targets of natural selection in the human genome[1]. However, the majority of studies carried out on modern human populations have largely been restricted to detecting selection events that have happened in the most recent periods of human history because selective sweeps decay due to processes including recombination and mutation[1] and can potentially be obscured by demographic events such as admixture[1,2]. By directly tracking genomic changes in aDNA, it may be possible to observe sweeps that otherwise are undetectable from modern data. However, until recently, the large sample sizes required for such analyses were unavailable and, as a result, many aDNA-based studies to examine natural selection were largely confined to specific alleles[3–7].

Increased sample sizes have enabled genome-wide selection scans on aDNA[4,5,8–13], but, most current approaches have focused on single-site statistics that leverage temporal data to detect allele frequency changes over time. An alternative strategy is to use haplotype-based approaches, which are sensitive to footprints of selection left behind by hitchhiking of linked alleles around an adaptive allele rising to high frequency[14–18]. However, most haplotype-based methods rely on phased genomes that are particularly challenging to obtain from ancient samples for several reasons. First, aDNA read lengths are extremely short (30–50 bp), and read-based phasing has reduced efficiency at these lengths[19]. Second, reference panels constructed from modern haplotypes may introduce bias in calling alleles from aDNA due to divergence that has arisen between ancient and modern genomes.

[1]Department of Integrative Biology, The University of Texas at Austin, Austin, TX, USA. [2]Department of Computational Medicine, University of California, Los Angeles, CA, USA. [3]Department of Ecology and Evolutionary Biology, University of California, Los Angeles, CA, USA. [4]Department of Human Genetics, University of California, Los Angeles, CA, USA. [5]Department of Statistics and Data Science, The University of Texas at Austin, Austin, TX, USA. [6]These authors contributed equally: Devansh Pandey, Mariana Harris. [7]These authors jointly supervised this work: Nandita R. Garud, Vagheesh M. Narasimhan ✉e-mail: ngarud@ucla.edu; vagheesh@utexas.edu

Recently, statistics that leverage multi-locus genotypes, which represent strings of unphased genotypes from diploid individuals, were proposed to circumvent the need for phased haplotypes[20–22]. However, a major challenge in applying these statistics to aDNA is its low coverage (largely between 0.5 and 2x coverage), which results in, on average, only one of the two diploid alleles being called[23]. In this study, we examined whether a previously described multi-locus genotype-based scan G12[22] might be suitable to detect adaptation in low-coverage aDNA data. In order to deal with biases in calling heterozygotes, the typical approach in aDNA is to utilize a pseudo-haploidization scheme[24], in which one allele per site is randomly selected to represent the genotype of the individual at that position. To evaluate the performance of this method on this type of data, we used simulations and examined well-characterized functionally validated variants, and compared its performance with another widely used selection statistic that can also be applied to unphased data. We then applied G12 to different epochs from an aDNA time transect to determine whether we could identify slices in time for which we see evidence for selection at well-characterized candidate sweeps. Finally, we examined novel targets of selection and assessed if our approach complements other studies of natural selection carried out using allele frequency-based methods[13].

We find that G12 has the power to detect a range of historical sweeps from aDNA and can recover several positive controls. Moreover, we find evidence for 14 candidate selective sweeps across the four time periods. However, some of the sweeps that are visible in the early periods of the European prehistory period are no longer visible in later periods. This is in agreement with previous work[2,13] made with allele frequency-based statistics showing that signatures of selection occurring several generations ago are undetectable due to the effects of recombination and mutation, or, from subsequent demographic changes including admixture.

## Results

### A time transect through Holocene Europe

In our analysis, we examined a collection of 708 recently published samples from Europe ranging from 6500 BP to 1345 BP (Supplementary Data 1)[8,25–36]. To minimize data processing issues across the set of samples, we included only samples for which hybridization capture was performed on 1.2 million positions[37], and that had at least 15,000 SNPs for which we could perform high-quality population genetic analysis. Additionally, we only included samples that did not appear to have significant contamination on the mtDNA or the X chromosome (in males) and were unrelated (up to the third degree). Finally, we only included samples that were uniformly treated with the same Uracil-DNA Glycosylase (UDG) process during library preparation and trimmed the last two bases from each read to reduce the impact of aDNA damage on our computed statistics (see Methods: aDNA data curation).

To homogenize the sample size of our analysis across time periods, we used 177 individuals for each epoch which we determined based on $f_4$-statistics, time period (based on direct radiocarbon dates or precisely dated archeological contexts), and geographic location (Fig. 1). Samples from each of these assigned population groups were genetically homogenous and had little to no ancestry from additional sources known to enter Europe and contributed in small proportions to a minority of European populations, including the Scythians and Sarmatians, the Uralic-related migrations into Hungary and Fennoscandia, and Iranian farmer related ancestry along the Mediterranean in Southern Europe. The groups of individuals were:

**N** First farmers of Europe from the Middle to Late Neolithic (abbreviated as the first letter of Neolithic). These individuals were from across Europe, are dated to between 6500 and 5019 BP, and are mixtures of European Hunter-Gatherer and Anatolian farmer ancestry.

**BA** Bronze Age Europeans (abbreviated as the first letters of Bronze Age). These individuals are from the Bell Beaker cultures of Western and Central Europe, dated between 4495 to 3808 BP.

**IA** Iron Age Europeans (abbreviated as the first letters of Iron Age). We used samples from Iron Age Britain and other countries in Western Europe dated between 3995 to 2350 BP.

**H** Finally, to represent Historical samples from Europe, we included samples from the Roman and late antique periods, primarily from Britain, dated from 2300 to 1345 BP.

### Detection of selection from aDNA with a multi-locus genotype statistic

G12, a multi-locus statistic analogous to haplotype homozygosity statistics, was recently developed to detect selection from unphased data[21,22]. On unphased diploid data, G12 is capable of detecting hard and soft selective sweeps, in which single versus multiple haplotypes bearing the adaptive allele, respectively, are at high frequency[38,39]. G12 is computed in windows comprising a fixed number of SNPs and is defined as:

$$G12 = (q_1 + q_2)^2 + q_3^2 + \ldots\ldots\ldots + q_n^2$$

where $q_1, q_2, q_3, \ldots\ldots, q_n$ denote the frequencies of unique n multisite genotypes, ranked from most common to most rare. The intuition behind this statistic is that haplotypes that have risen to high frequency are present in a homozygous state also at high frequency[22].

However, the low coverage (mean of 1.5×) of aDNA data means heterozygote sites are unable to be called effectively[40–43]. To address this issue, we process aDNA samples as "pseudo-haploid" data where one read mapping to a position is chosen at random, and the allele of that read is then used as the genotype of the resulting "pseudo-haplotype"[24]. Thus, in our application of G12, $q_1, q_2, q_3, \ldots\ldots, q_n$ denote the frequencies of unique n multisite pseudo-haploid genotypes. Additionally, to evaluate the effects of aDNA degradation and missing SNPs on G12, we mimic aDNA damage, the most significant of which is deamination[44] (Supplementary Figs. 1, 2, Methods: Generation of modern human data mimicking ancient data, Supplementary Table 1 and Supplementary Fig. 5).

Below, we test the ability of G12 in simulated data to distinguish between selection and drift in a range of simulated evolutionary scenarios. Furthermore, we demonstrate the ability of G12 to identify well-characterized and functionally validated variants that have previously been found to be under selection by multiple modern and ancient genomic studies[3,5,8,13,34,45] (Supplementary Table 2). In both types of tests, we compare the performance of our method with another approach SweepFinder2 (SF2)[46], which also uses local multi-locus statistics and has been recently used in a selection study using aDNA[2,46].

### Evaluating G12 on simulated data

To evaluate the performance of G12 on simulated aDNA data, we used the forward-in-time simulator SLiM 3[47] to generate genotypes incorporating missingness, ascertainment bias, random allele calling, and genotyping error that are typical of the aDNA data used in our study (Methods: Generation of simulated data). We simulated hard and soft sweeps in a population under the Tennessen et al. model[48], a demographic model that captures broad features of the allele frequency spectrum of modern European genomes.

In a hard sweep, a single haplotype carrying the adaptive mutation increases in frequency within the population. By contrast, in a soft sweep, multiple haplotypes rise in frequency simultaneously. Both hard and soft sweeps can originate from de novo mutations or from standing genetic variation (SGV). In the de novo case, a hard sweep is likely if the adaptive mutation rate is low ($\theta_A = 4N_e \mu_A << 1$, where $N_e$ is

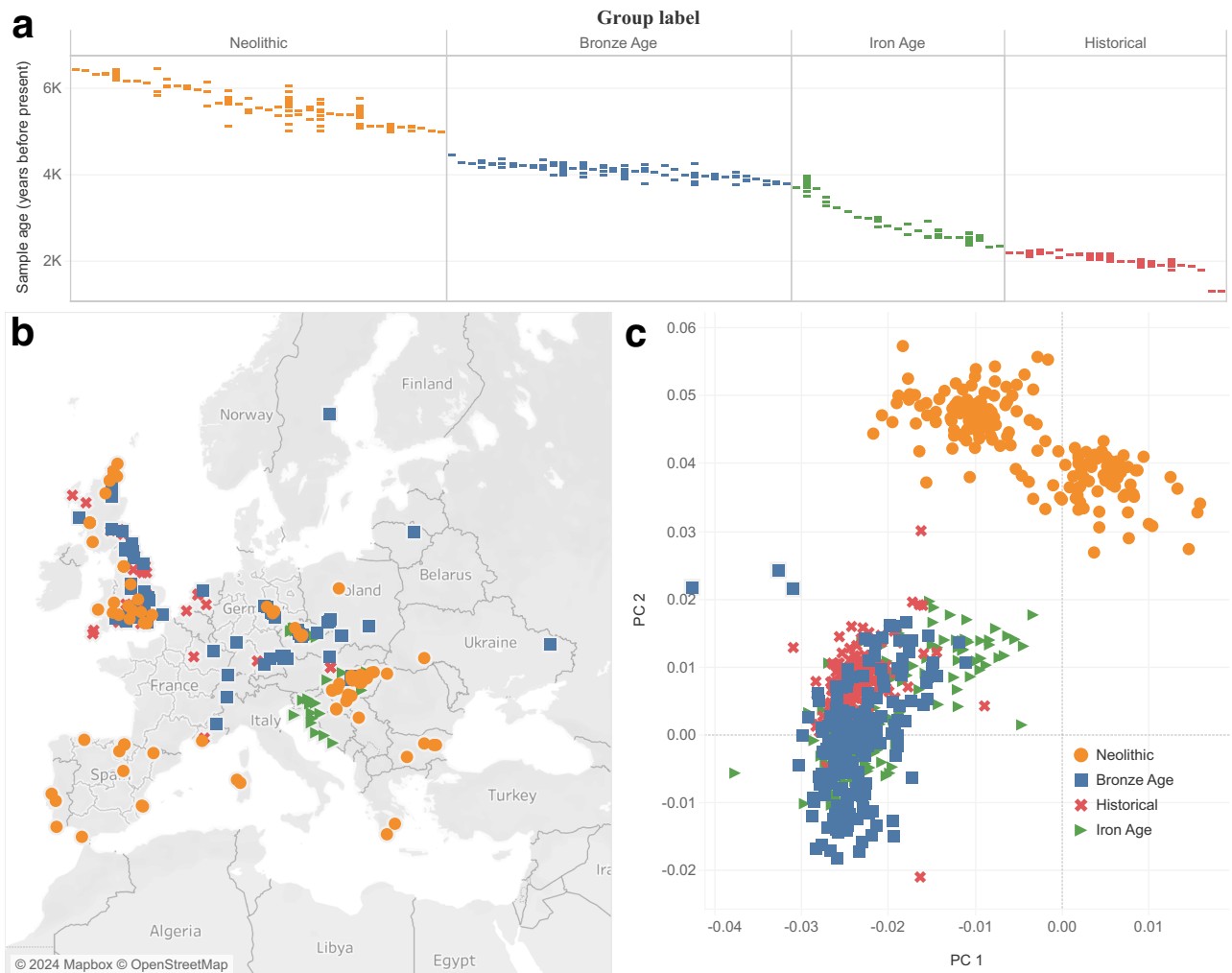

**Fig. 1 | aDNA samples included in this study. a** Distribution of archeological or radiocarbon dates for sites (vertical columns) from each time period over the past 7000 years. Each colored bar represents single samples from a site that has been dated to a particular time. Multiple samples from the same site are annotated along the same column. **b** Locations of ancient individuals that passed our analysis thresholds, forming a sample size of 708 individuals. **c** PCA analysis of ancient individuals projected onto a basis of modern samples. PC 1 principal component 1, PC 2 principal component 2.

the effective population size and $\mu_A$ is the mutation rate of the adaptive mutation), whereas a soft sweep is expected when it is high ($\theta_A > 1$)[38]. When sweeps arise from the SGV, if the initial partial frequency before the onset of selection is low, the adaptive allele is more likely to be on a single genetic background, resulting in a hard sweep. However, if the initial partial frequency before the onset of selection is high enough for the adaptive allele to be present on multiple distinct haplotypes, a soft sweep is expected.

To simulate hard sweeps, we introduced a single mutation at the center of a 500KB chromosome and allowed it to increase in frequency until the time of sampling. In addition, we simulated sweeps from recurrent de novo mutations for $\theta_A = 1$ and $\theta_A = 10$ (Methods: Running selection scans on simulated data). When $\theta_A = 1$, 77–97% of simulations resulted in soft sweeps, while for $\theta_A = 10$, all simulations resulted in soft sweeps (Supplementary Fig. 6). We also simulated sweeps from SGV with initial allele frequencies of $f_{init} = 0.001$, 0.005, and 0.01 (see Methods: Running selection scans on simulated data). When $f_{init} = 0.001$, only 5–29% of simulations had two or more distinct haplotypes bearing the adaptive mutation at the onset of selection. This proportion increased to 51–88% when $f_{init} = 0.005$, and to over 92% when $f_{init} = 0.01$ (Supplementary Fig. 7).

To mimic the epochs in which selection may have started and the time of sample collection, we varied the time of the onset of selection

and the time of sampling, spanning the past ~7000 years (250, 100, and 40 generations before present). We note that in this simulation setup, selection never ceases once turned on. We obtained three samples of 177 individuals matching the sample size of our dataset. Once we had generated our aDNA data, we ran G12 using a fixed window of 201 SNPs, corresponding to roughly 490,000 base pairs.

We first evaluated the impact of pseudo-haploidization on the performance of G12. We found that G12 yields similar results in both pseudo-haplotype and MLG data (Supplementary Fig. 8). In fact, G12 calculated on pseudo-haploid data from older and stronger sweeps shows an increase in power compared to G12 calculated on MLG data. This is likely due to a higher probability of detecting haplotypes at a high frequency when clustering pseudo-haplotypes with exclusively major and minor alleles, as opposed to MLG's with major-homozygous, minor-homozygous and heterozygote genotypes.

Next, we incorporated missing data at levels typically observed in aDNA data. Missing data can lead to artificial inflation of the G12 statistic and, therefore, diminish power due to the clustering of haplotypes that match at all non-missing positions (see Methods: Running selection scans on simulated data). To mitigate this effect, we adjusted our haplotype clustering scheme so that haplotypes with more than 90% of missing data would not be clustered with another haplotype group and instead are treated as unique haplotypes in our

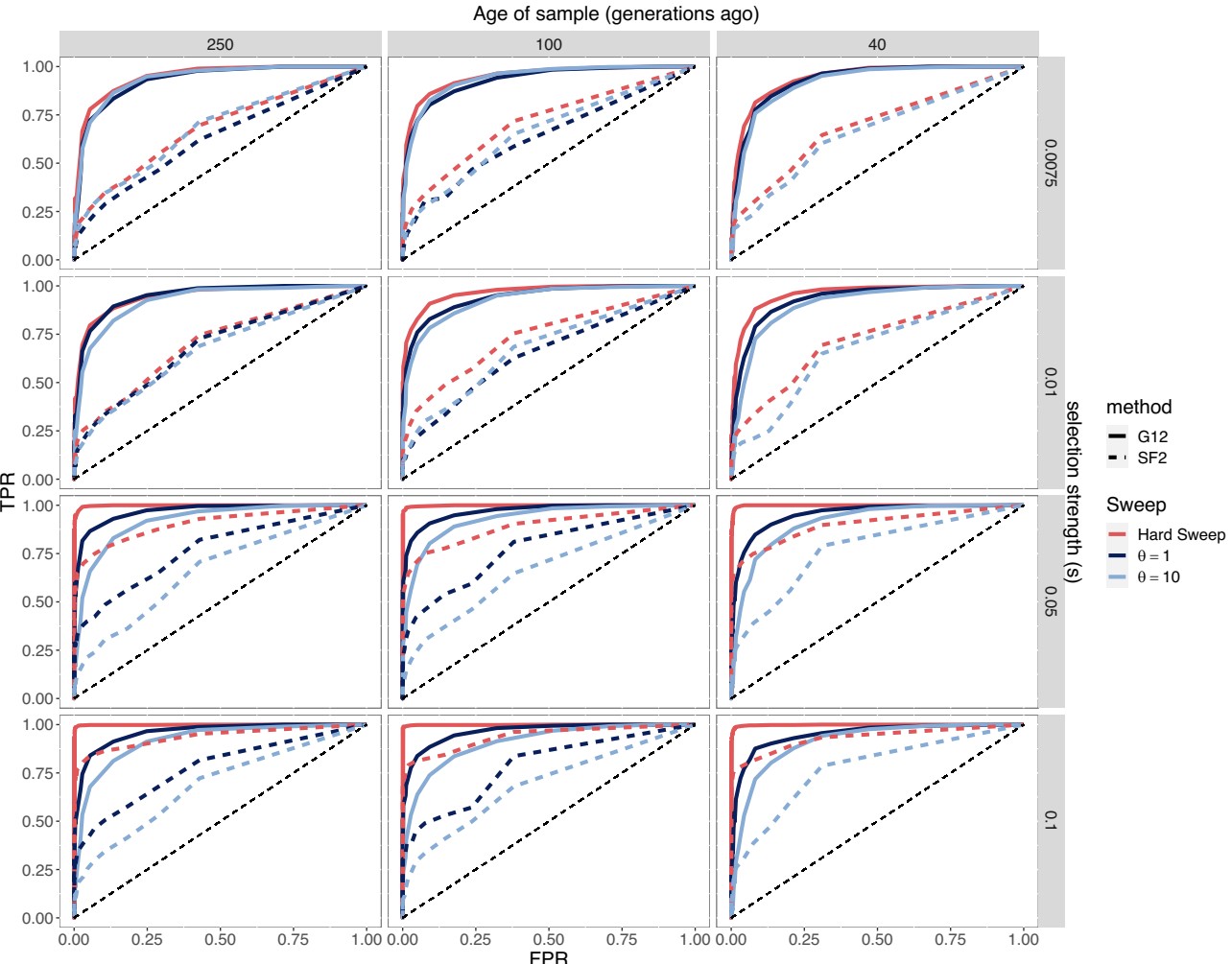

**Fig. 2 | Receiver operating characteristic (ROC) curves of G12 and SF2 in detecting single origin hard sweeps (red) versus sweeps from recurrent de novo mutations (blue) in simulated aDNA data.** Each panel shows ROC curves for G12 and SF2 for varying strengths of selection (rows) and age of the sample (columns). For the simulations considered here, the onset of selection was set to 1000 generations before present. In Supplementary Fig. 10, both the age of the sample and the onset of selection are varied, and Supplementary Fig. 11 shows the ROC curves for sweeps from SGV. We computed G12 and the SF2 CLR scores in a total of 1500 simulations (500 hard sweeps, 500 soft sweeps, and 500 neutral simulations) for each combination of parameters with mutation rate μ = 1.25 × 10⁻⁸/bp, chromosome length L = 5 × 10⁵ and recombination r = 1 × 10⁻⁸ events/bp. TPR true positive rate, FPR false positive rate.

sample. This approach was particularly effective for improving the ability of detecting older sweeps (onset of selection 1000 generations ago), yielding similar power as G12 in the case of no missing data (see Methods: Running selection scans on simulated data; Supplementary Fig. 9).

Next, we evaluated the power of G12 in detecting hard and soft sweeps arising from both recurrent de novo mutations and SGV in aDNA simulated data. Our results show that G12 has reasonable power to detect both hard and soft sweeps, albeit with greater power to detect hard sweeps (Fig. 2 and Supplementary Figs. 10, 11). The footprint of a hard selective sweep is expected to extend over $s/[\log(N_e s) r]^{49}$ base pairs, where $s$ is the selection strength, $N_e$ the effective population size and $r$ the recombination rate. On average, 201 SNP windows correspond to ~490,000 bp. With such a large window size and a recombination rate of $r = 1 \times 10^{-8}$, we should expect to be able to detect sweeps with selection strength $s = 0.04$ or greater. Weaker sweeps could be detected by reducing the window size used to compute G12 at the expense of some peaks being false positives due to random fluctuations in homozygosity in the data. Given the noisy nature of our data, we decided to use a conservatively large window size to avoid detecting false signatures of sweeps.

Finally, we compared G12's performance against that of SF2[46], an SFS-based approach for sweep detection that has been applied previously on aDNA[2]. SF2 uses a composite-likelihood ratio test (CLR) that looks for characteristic skews in allele frequencies that are expected under hard sweeps relative to neutrality. Our results using G12 on pseudo-haploid data show a substantial improvement in the detection of both hard and soft sweeps compared to SF2 (Fig. 2 and Supplementary Fig. 11 showing substantial differences in AUC and Supplementary Fig. 10 showing differences in power curves), indicating the improved ability of G12 in detecting sweeps compared with SF2.

## Application of G12 to functionally validated variants from real data

To test the ability of G12 to detect selection signals on real data, we modified modern genomic data from European individuals from the 1000 genomes project[50], by introducing missingness, ascertainment bias, aDNA damage, and pseudo-haploidization to mimic aDNA data (Methods: Generation of modern human data mimicking ancient data).

First, we examined the correlation between G12 values computed genome-wide on diploid low-coverage data from the 1000 genomes project[42] without our aDNA mimicking process to that of G12 values

computed on the same samples but using our aDNA mimicking process (Methods: Running selection scans on ancient datasets and Supplementary Fig. 12). The correlation between the results across all windows in the genome was 0.95 suggesting that the performance of G12 on aDNA data would likely be highly similar to diploid data (Methods: Running selection scans on ancient datasets and Supplementary Fig. 12).

However, our high genome-wide correlations might not translate to signals of detectable positive selection at the tails of the distribution. Therefore, we examined the ability of G12 to detect classic selective signals in the genes *LCT*, *TLR1*, and *SLC24A5*, which have been identified by multiple previously conducted selection scans and are regions that are highly differentiated between Europeans and Asians (Supplementary Table 2). The causal alleles at these loci have been fine-mapped in association studies and have also been functionally validated in cellular assays. These alleles are commonly used as positive controls in studies carrying out tests for natural selection in humans[45]. The *LCT* locus is responsible for conferring lactase persistence into adulthood; *TLR1* is a gene involved in immune cell response and *SLC24A5* is the dominant locus contributing to light skin pigmentation in Europeans[3,51].

Using our aDNA mimicking process on the modern data and then applying G12, we were able to identify peaks that included the functionally validated causal allele *LCT*, *SLC24A5*, and *TLR1* in the top peaks observed chromosome-wide in the European (CEU) population but not in African (YRI) and South Asian (STU) populations (Fig. 3a). We also examined the effect of utilizing different parameters for window-sizes and jumps (distance between analysis windows in terms of number of SNPs) and obtained an optimal choice of these parameters on real data (see Methods: G12 parameter choices and peak calling and Supplementary Fig. 14). To compare the performance of G12 with SF2, we also ran SF2 on this modern data mimicking aDNA data. To detect genes that were significant, we followed an approach previously used by ref. 2 which used SF2 to examine an aDNA data using the same ascertainment scheme as our current dataset, keeping our parameter choices identical (Methods: Running selection scans on modern data mimicking aDNA). After obtaining CLR scores for CEU, YRI, and STU populations, we applied the outlier gene detection methodology as described in ref. 2 to identify regions of the genome that were under selection. Using this method, we did not find CLR score outliers that overlap the *LCT*, *TLR*, and *SLC24A5* genes in the CEU population, either by considering the 50 kb flanking region as in ref. 2 or extending this to 265 kb as done with the G12 scan. While there might be a gene outlier close to the *LCT* locus, this peak is multiple hundreds of thousands of megabases away from the *LCT* gene and is not the highest peak chromosome or genome-wide (unlike with our multi-locus genotype approach). We report all outlier genes considered to be significant using the Souilmi et al.[2] approach for the YRI, STU, and CEU populations using SF2 in Supplementary Data 1. These results from SF2 contrast with our approach, where we identify these positive controls within windows that are amongst the top three genome-wide (Methods: Running selection scans on modern data mimicking aDNA).

Next, to establish that our process could identify the timing of signals of natural selection from aDNA, we examined the *LCT* locus at different time periods of European history. This locus is particularly relevant for this analysis as the causal allele was absent in Europe prior to the arrival of Steppe Pastoralists in the Bronze Age and, therefore, could not have been under selection prior to that point[3,5,8,12,13,45,52–54]. By applying G12 across different periods in our time transect, we show that we were able to identify selection at the *LCT* locus in the H period (this window is the top peak genome-wide), but we do not see signals of selection for these in the Bronze Age and Iron Age populations (Fig. 3b). These results, therefore, are in line with the rapid increase in frequency of the causal variant rs4988235 only in the H period (Fig. 3c), a finding that has also been replicated in other aDNA studies[8,13,45].

Taken together, our results suggest that G12 performs significantly better than SF2 in simulations and at identifying well-characterized selective signals, in line with the theory that SF2 was primarily designed to detect hard sweeps that are fixed.

## Selective events in earlier periods are obscured by admixture or drift

Having established that G12 can identify signals of selection in simulated data and correctly distinguish between positive and negative controls in modern data mimicking aDNA, we next applied G12 to our aDNA time transect. To identify analysis windows that have exceptionally high G12 values that are unlikely to be generated under neutrality, we defined a genome-wide threshold for significance as the fifth highest G12 value obtained from 58,350 neutral simulations, which equals 10× the number of independent analysis windows in the G12 scan[48] (see Methods: Running selection scans on simulated data, G12 parameter choices, and peak calling). G12 values above this threshold were classified as putative sweeps. As windows adjacent to each other may belong to the same selective sweep, consecutive analysis windows above the G12 neutral threshold were assigned to a single selective event, or "peak". The highest G12 value among all windows of a peak was chosen to represent the value for the whole peak. To remove spurious peaks that might have arisen due to high rates of missing data or low recombination rates, we masked the peaks located in those regions (see Methods: Quality control for removing false sweeps). We used a window size of 201 SNPs as this was the optimal choice from our ancient DNA mimicking process. At this threshold, approximately five to six candidate peaks (top 0.1% of windows) were identified per epoch that reached significance at the genome-wide level.

We began by re-examining 12 loci previously established to be under selection using aDNA data[8]. Although the selection signals produced by this previous scan and our scan differ in their methodology and, therefore, their ability to detect selective events, we wanted to assess if we might be able to use our approach to localize when in time these signals were selected.

As seen in Fig. 4a, the time period in which we observe a signal of selection at the *LCT* locus is limited to the H period. In the N population, among the top peaks, we found a signal which included the gene *OCA2/HERC2*, variations in this gene are associated with eye, skin, and hair pigmentation variation[8,25,55]. This gene is the primary determinant of light eye color in Europeans. The earliest people in the anthropological record to have light eye pigmentation are European Hunter-Gatherers. Our analysis suggests selection for light eye pigmentation in farming populations from Anatolia with dark eye pigmentation who admixed with hunter-gatherers during their expansion into Europe[13]. In addition, we also observe a signal of selection at the *HLA* and at neighboring *ZKSCAN3* in most of the epochs (Fig. 4a, c).

Outside of these four loci, our selection scan also revealed several other candidates which we determined as being above our significance threshold. Several of these were associated with skin and eye pigmentation. In the BA and IA epochs we observed a signal of selection in the gene *SCL24A5*. As mentioned above, this gene is thought to be the major determining locus for light skin pigmentation in Europeans[3,56]. While highly differentiated between Asians and Europeans and appreciated as a major candidate of selection using modern European Genomes, single SNP allele frequency approaches examining aDNA have yet to identify this particular allele as a candidate[8,13]. This shows the value of employing alternate types of selection scans on similar datasets to uncover putative selective sweeps.

We observed a signal at a locus associated with *PPM1L* as one of the top peaks in N, which is an obesity related marker in humans[57]. This signal for selection on obesity and body weight related alleles during the Neolithic or the change in dietary practices from hunting and gathering to farming is also observed in single SNP based approaches[13].

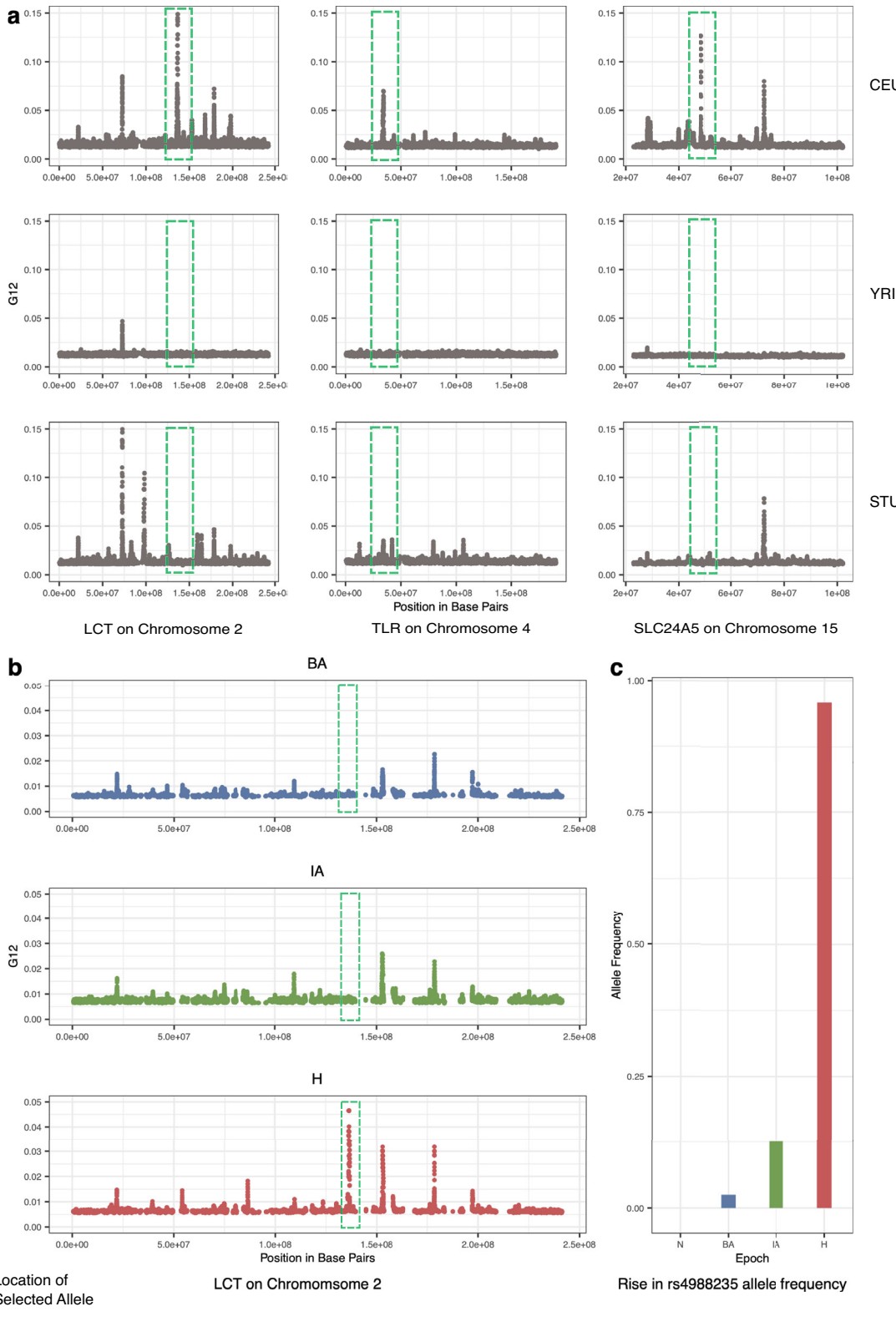

**Fig. 3 | Recovery of variants well characterized to be under selection in modern Europeans (positive controls). a** G12 values for modern population data from the 1000 genomes project[50], which was modified to mimic aDNA. G12 can detect several variants that have been previously found to be under selection in modern Europeans. However, these signals are completely absent or highly attenuated in populations of other ancestries (YRI and STU). **b** On the aDNA time transect, the rs4988235 locus in the *LCT* gene is the top peak genome-wide in the H population but is absent in BA and IA populations. **c** The allele frequency reaches near fixation in the H population but is at considerably lower frequency in BA and IA periods as it was only introduced into Europe by the arrival of pastoralists from the Pontic-Caspian Steppe[13]. In panel (**b**) we show that we observe high G12 values only in the H period but not in previous time periods as a demonstration of our ability to localize the timing of selection to various epochs.

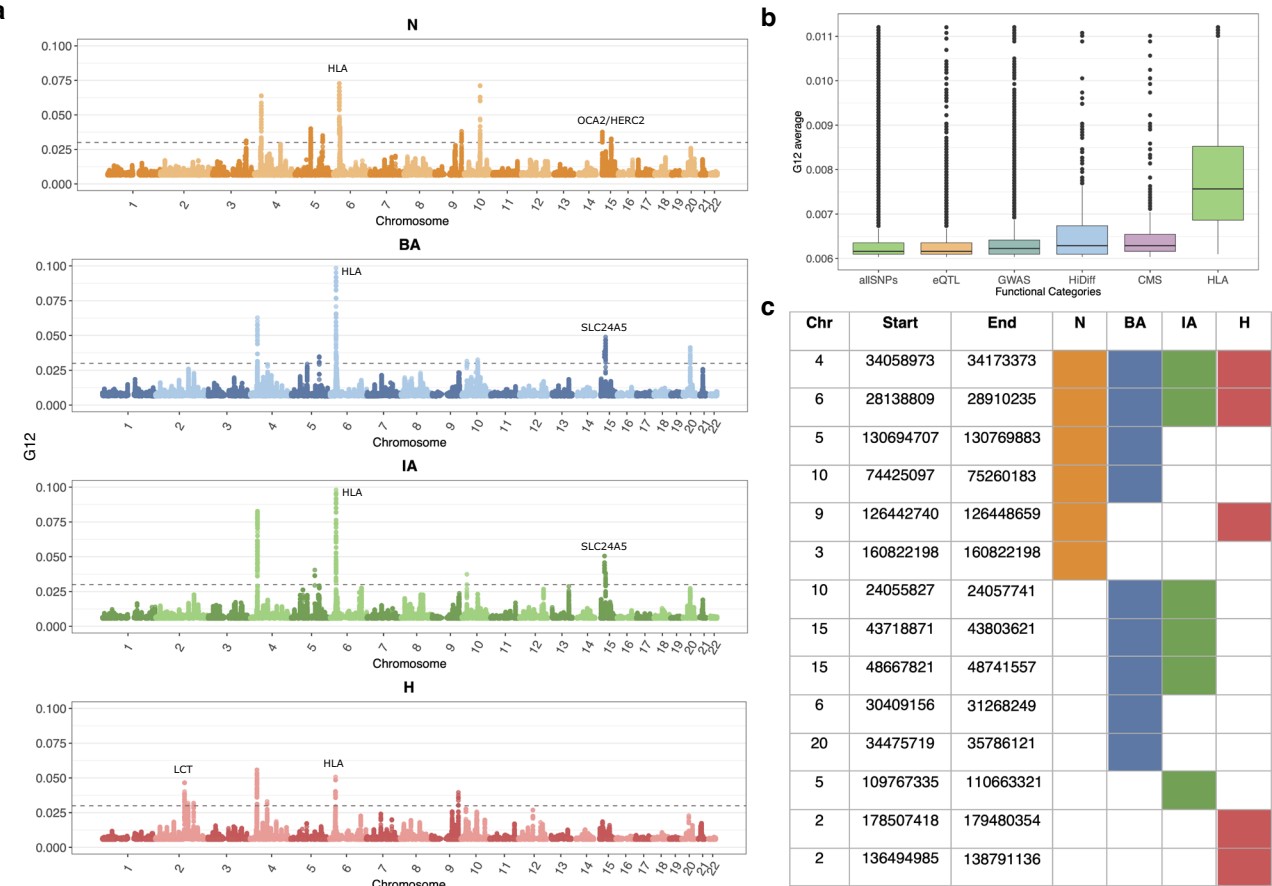

**Fig. 4 | G12 applied on aDNA time transect data. a** Manhattan plot of G12 values genome-wide with some signals with known functional consequences identified by earlier scans annotated. The gray dashed line is the genome-wide significance threshold based on simulations under the Tennessen et al. demographic scenario[48]. **b** Box plots represent the distribution of the G12 average across different functional categories. The Center line represents the median. Boxes extend from the 25th to 75th percentile (interquartile range, IQR). Whiskers extend to the minimum and maximum values, excluding outliers. Outliers are shown as individual points. Sample size ($n$): allSNPs = 726,922, eQTL = 8002, GWAS = 8986, HiDiff = 699, CMS = 710, and HLA = 3817. Sample size ($n$) reflects the number of biological replicates, as they represent independent positions in the genome across different individuals. **c** Unique peaks across each epoch. We show peaks that are shared across epochs and also the peaks that are localized in a single epoch. Chr, Start, and End refers to the chromosome, start, and end position of each peak, respectively.

In addition, we observed several signals in genes that were associated with immunity or auto-immunity. In the BA population, we observed a candidate in the locus containing *ADK*, which regulates the intra and extracellular concentrations of adenosine which has widespread effects on cardiovascular and immune systems[58,59]. We see a signal at the *ITCH* gene in the BA, which is associated with immune response, and regulation of autoimmune diseases[60,61]. In the IA, we see candidate sweep at the *MAN2A1* locus—genetic variations in this gene have been shown to cause human systemic lupus erythematosus[62]. In Fig. 4c, we report a list of all regions that appear to be under selection in each epoch, along with a full list of genes mapped at each peak by VEP[63] in Supplementary Data 1.

We next examined the overlap between selective signals across time periods. In doing so, we found that few signals, such as the HLA locus, were detected in all four time periods. Of the 14 unique regions that we identified as signals of selection, 11 out of 14 were seen either only in one time period or the adjacent time period and only half of the signals we observed in the earliest time were also seen in the latest time period. Our results are consistent with two other approaches using complementary selection statistics that also observed only limited overlap in signals across time periods. Namely, ref. 13, which utilized differences in allele frequencies before and after admixture at a single locus, and ref. 2 which utilized SF2. Taken together with these studies, our results suggests that either neutral processes such as drift

(encompassing mutation, recombination and population size changes) or admixture in the several thousand years between the earlier and later time periods could have obscured the selective sweep signatures.

## Gene set enrichment analysis

In addition to examining individual SNPs, we examined mean G12 values across broad categories of functional SNPs. We looked at loci that were associated with changes in gene expression (eQTLs), identified as associated in genome-wide association studies (GWAS), or were thought to be previously under selection in Europeans (CMS) or highly differentiated between Europeans and Asians (HiDiff) or were part of the HLA region. We found that functional categories of SNPs were seen with significantly higher G12 values compared to SNPs that were not annotated as being functionally relevant (allSNPs), with the HLA region being the most elevated of the functional categories (Fig. 4b).

We next asked if we could associate biological functions with these top-scoring loci. We computed *p* values based on deviation from neutrality based on simulations (see Methods: Enrichment analysis). To determine if categories of genes associated with genome-wide association studies were significantly associated with selection signals, we carried out enrichment analysis using FUMA[64], which maps SNPs to genes and performs gene set enrichment analysis for GWAS annotations incorporating LD information as well as gene matching by length

and conservation scores (see Methods: Enrichment analysis). We found that many categories of GWAS related to anthropometric traits, autoimmune traits as well as disease-related traits were under selection across the different time epochs. We report gene sets from the GWAS catalog using FUMA: Gene2Func[64] and used a conservative significance threshold of $-\log_{10} p \geq 5$. We report the list of all categories for which we observed enrichment in Supplementary Fig. 16.

## Discussion

In this paper, we showed that a previously described selection statistic that is designed for unphased diploid data[16,22] can be applied to aDNA data. To date, while allele frequency-based approaches have been used extensively in the field, approaches using haplotype scans have largely been lacking. A single study[65] performed a selection scan by phasing low-coverage aDNA samples, and running a widely used extended haplotype statistic, XP-EHH. But the effect of phasing and imputation, as well as the performance of this statistic on such data were not fully evaluated. Here, we took a conservative approach aimed to reduce bias and artifacts from the use of modern reference panels for phasing and imputing low-coverage ancient DNA. However, we show that we are able to maintain the power to detect selective signals in simulations and recover exemplar variants thought to be under selection in recent history.

Our results, which take advantage of the major increases in sample size in the availability of aDNA data in the past 5–10 years, demonstrate the potential of running multi-locus genotype-based scans on aDNA. G12, can potentially be employed in other settings where sequencing coverage is low and there is high missingness requiring pseudo-haploidization. Importantly, since haplotype-based statistics are not as reliant on temporal data to exclude false positives, these statistics are useful for ancient datasets from geographic regions that only have a single timepoint.

Despite its potential, our approach also has several limitations. As the results from the simulation study show, G12 is powered mostly for strong selective sweeps ($s > 0.01$), biasing our results towards the more extreme and potentially interesting selective events in human history. Moreover, the timing of the onset of selection is limited by our ability to detect selection below this high threshold, and therefore, lack of selection at a particular time could also be due to a lack of power. Another major limitation of our approach is that our window-based method is unable to localize selection to a specific allele, though few approaches currently are capable of doing so. On data from a capture array like we have, this distance can span large distances and decrease our target resolution. Here we used the closest gene to the peak SNP in a series of windows to connect genes to candidates under selection.

Our benchmarking results on simulated data show that G12 is a better statistic than SF2 on pseudo-haploid aDNA data. In addition, we showed that, unlike G12, the SF2 statistic was unable to pick up on well-characterized sweeps in historical and modern European samples. In addition, on application to real data, we show that a significant fraction of the regions identified by SF2 lie in regions of low recombination or high data missingness, perhaps suggesting that such QC steps are also necessary in applying that statistic to aDNA data, which has elevated error rates and high missingness.

In addition, to highlighting the robustness of the G12 statistic on this type of data, our work now adds an additional complementary line of evidence for a lack of overlap between signals identified from time periods several thousand years in the past and modern times. This could be due to either the effects of recombination, mutation, and drift eroding a signal that had occurred far in the past in the several thousand-year gap between our earliest and latest samples, or it could be that large-scale population turnover (up to 90% admixture), which occurred in Europe during the Neolithic and again in the Bronze Age has greatly reduced the frequency of such haplotypes in the population. While our method does not allow us to distinguish between these

scenarios, we note that this observation is unlikely to be due to differential power across time periods to detect selective signals because our sample sizes, missingness, FDR-based thresholds, and other computed genetic diversity parameters do not appear to be significantly different from one time period to another.

An important future direction for this type of research is to carefully examine the accuracy of imputation and phasing on low-coverage ancient data using biological confirmation such as from trios, which have become available in small sample sizes in aDNA now. Future directions could also be to extend these scans to other time periods or, more importantly, to other geographic regions in the world where aDNA data were becoming rapidly available to fully understand the dynamic nature of adaptation over the course of human history.

## Methods

### aDNA data curation

We obtained aDNA data from Allen Ancient DNA Resource[66] (https://reich.hms.harvard.edu/allen-ancient-dna-resource-aadr-downloadable-genotypes-present-day-and-683, version 51), and selected the samples that were enriched for 1240k nuclear targets with an in-solution hybridization capture reagent. We did not include individuals if they had less than a 3% cytosine-to-thymine substitution rate in the first nucleotide for a UDG-treated library, as these were indications of contamination. We also removed individuals who had indications of contamination based on polymorphism in mitochondrial DNA or the X chromosome in males, based on estimates from contamix[67] and ANGSD[68]. For population genetic analysis to represent each individual at each SNP position, we randomly selected a single sequence (if at least one was available).

Finally, we assembled genome-wide data of various human populations from Holocene Europe dated between ~7000 BP and 1300 BP. To maintain homogeneity across time periods, we sampled 177 individuals from each archeological period—the Neolithic, the Bronze Age, the Iron Age, and the Historical period. For populations with more than 177 individuals, we only chose samples from these periods with the highest coverage and prioritized samples from the same site whenever possible. A list of all samples analyzed is in Supplementary Data 1.

Within each time period, in selecting our subset of 177 samples to use for each time period carried out a qpAdm analysis (which utilize different combinations of pairwise f-statistics) to remove individuals with ancestry atypical of that time period in ancient Europe[29]. Specifically, we removed individuals who could not be modeled by a mixture of Anatolian Farmers, European Hunter-Gatherers, and Steppe Pastoralists who have made the largest genetic contribution to modern Europeans. This ensured that our samples did not contain elevated levels of Iranian farmer, or East Siberian Hunter-Gatherer, or African ancestry known to admix into some but not all Europeans.

### Principal components analysis

We carried out PCA using the smartpca package of EIGENSOFT 7.2.1106[69,70]. We used default parameters and added two options (lsqproject: YES, and numoutlieriter:0) to project the ancient individuals onto the PCA space. We used 991 present-day West Eurasians as a basis for the projection of the ancient[29,71]. We restricted these analyses to the dataset obtained by merging our aDNA data with the modern DNA data on the Human Origins array and restricted it to 597,573 SNPs. We treated positions where we did not have sequence data as missing genotypes.

### Generation of modern human data mimicking ancient data

To examine whether the G12 based selection scans would be applicable to aDNA data; we developed a process of converting the modern human genomic data from the 1000 Genomes project[50] to mimic the statistical and physical properties of aDNA data and ran the scans on modified modern data. We utilized a pseudo-haploidization scheme[24]

in which we randomly selected one of the reads mapping a position and assigned the sample the genotype of that read as described in Supplementary Fig. 1.

To simulate ascertainment, we restricted the 1000 genome samples to just the 1.2 million positions that were on the aDNA capture array. Finally, we incorporated missingness on a per-site basis in modern data using the mean (0.55) and standard error (0.23) we observed in our sample of 708 individuals and randomly set the genotypes of a certain proportion of individuals in the modern data to missing (Supplementary Table 1 and Supplementary Fig. 2).

We also introduced damage at rates typical to aDNA in modern data from 1000 genomes project[50]. We incorporated damages caused by deamination in aDNA, which typically results in substitution of C to T nucleotide in aDNA[44], we modified C alleles to T alleles every 100 positions in each individual.

## Heterogeneity in levels of missingness in the aDNA data

Before running the G12 selection scan on aDNA, we wanted to examine whether missingness in the data was non-random. This would, in turn, influence our simulations and our testing on modern data. First, we examined the average number of missing genotypes per SNP per window and found that the missingness in our data was normally distributed (Supplementary Fig. 3), suggesting that missingness could be generated from a random process, as one would expect if degradation of DNA molecules across different sections of the genome were approximately the same.

To directly evaluate region-specific missingness, we computed positional autocorrelation for the fraction of missing individuals in SNPs that are adjacent to each other, found out that the autocorrelation between the fraction of missingness individuals was very low (less than 0.2) and did not have significant change in its correlation beyond correlation seen between SNPs that are two SNPs apart (Supplementary Fig. 4). This suggests that missingness across positions is not correlated.

We then added missingness to our modern ancient DNA data using the actual levels of autocorrelation in the mean and variance of fraction of missing individuals as well as adding missingness without adding autocorrelation and found that the distribution of G12 values across a chromosome were nearly identical to one another (Supplementary Fig. 5) in line with the idea that missingness is largely random, and modeling low levels of positional autocorrelation does not lead to significant differences in the distribution of G12 values. Finally, to be extra conservative, we removed windows which had high levels of missingness from the data at thresholds indicated by (Supplementary Fig. 3).

## Generation of simulated data

We simulated all our genotype data using SLiM 3.7[47,48]. For all our simulations, we use a mutation rate $\mu = 1.25 \times 10^{-8}$/bp, chromosome length $L = 5 \times 10^5$, and recombination $r = 1 \times 10^{-8}$ events/bp. To achieve an equilibrium level of genetic diversity, we added a neutral burn-in period of $10 N_{e,\text{ancestral}}$ to our simulations, where $N_{e,\text{ancestral}}$ is the ancestral effective population size.

To simulate hard sweeps, we introduced a single beneficial mutation at the center of the simulated chromosome. We also simulate sweeps arising from recurrent de novo mutations and from SGV. To simulate sweeps from recurrent de novo mutations, we introduced adaptive mutations at a rate defined by for $\theta_A = 1$ and $\theta_A = 10$, where $\theta_A = 4 N_{e,t} \mu_A$ with $N_{e,t}$ the effective population size at generation $t$ and $\mu_A$ is the mutation rate of the adaptive mutation. To simulate sweeps from the SGV we introduced a neutral mutation to the center of the chromosome and let it rise in frequency until it reached a partial frequency of $f_{\text{init}} = 0.001$, 0.005, or 0.01, when the mutation became beneficial. We conditioned all simulations on the presence of the adaptive mutation, that is, we restarted the simulation if the adaptive

mutation was lost. We varied the time at which the mutations were introduced, $t = 280$, 500, and 1000 generations ago, along with their selection coefficient ($s$), and sampled the population at three different time points: 250, 100, and 40 generations before present. For all simulations, we pseudo-haploidized[40–43] the data as described above (Supplementary Fig. 1), and to incorporate the sparsity of aDNA data, we randomly selected 201 SNPs. That is, for each simulation, we obtained a 201 SNP window for our sample of 177 individuals.

Both hard and soft sweeps can occur in the case of sweeps from recurrent de novo mutations and sweeps from the SGV. To understand the likelihood of hard versus soft sweeps in the recurrent de novo mutations scenario, we computed the number of independent mutational origins of the adaptive mutation at the time of sampling across all simulations. Two or more independent origins indicate soft sweeps, while a single origin indicates a hard sweep. We found that for $\theta_A = 1$, 77–97% of simulations resulted in soft sweeps, while for $\theta_A = 10$, all simulations resulted in soft sweeps (Supplementary Fig. 6) To understand the likelihood of soft versus hard sweeps in the SGV case, we computed the number of distinct haplotypes bearing the adaptive mutation before the onset of selection across all simulations. We found that when $f_{\text{init}} = 0.001$, most sweeps were hard with only 5–29% of simulations with two or more distinct haplotypes bearing the adaptive mutation. This proportion increased to 51–88% when $f_{\text{init}} = 0.005$, and to over 92% when $f_{\text{init}} = 0.01$ (Supplementary Fig. 7). In cases where the mutation was introduced more recently (280 generations ago) we observed a higher proportion of soft sweeps due to the higher $N_e$ of that period. This higher $N_e$ means that there will be a higher number of haplotypes bearing the adaptive mutation for each of the $f_{\text{init}}$, this increases the number of distinct haplotypes before the onset of selection, increasing the number of soft sweeps.

## Running selection scans on simulated data

We computed G12 in simulated data using 201 SNP windows in 500 simulations for each combination of parameters tested. We first obtained G12 for hard sweeps and neutrality ($s = 0$), with and without applying our pseudo-haploidization scheme and with no missing data (Supplementary Fig. 8).

Based on the missingness observed in our aDNA data, we added missing data to our simulated datasets following a beta distribution with a mean of 0.55 per SNP and a standard deviation of 0.23 (Supplementary Table 1). Elevated levels of missing data can artificially inflate G12. This increase of G12 with elevated levels of missing data stems from the original clustering scheme used to compute the statistic. Previously, a missing allele in a haplotype within an analysis window would be ignored and the haplotype would be clustered with another haplotype matching at all other non-missing sites. High levels of missing data would lead haplotypes with high missingness to cluster together as a single haplotype, creating a haplotype at high frequency in the population and consequently inflating G12. To mitigate this effect, we adjusted the clustering scheme such that if a haplotype within the window had more than m% of missing data, then we would not group it with another haplotype, and instead treat it as a singleton. In our work, we set this threshold to 90%, that is, if a haplotype had more than 90% missing data, then it was treated as a singleton. Supplementary Fig. 9 shows the power of G12 in three scenarios: no missing data, missing data, and missing data, but adding the adjusted scheme with a threshold of 90%.

Additionally, we tested the ability of G12 to detect hard and soft sweeps arising from recurrent de novo mutations and from the SGV (Supplementary Figs. 6, 7). In the case of sweeps from de novo mutations we observed that G12 has greater power to detect hard sweeps. This power is reduced the softer the sweep is ($\theta_A = 10$; Fig. 2 and Supplementary Fig. 10) or if the sweep is young and does not yet have multiple haplotypes at an appreciable frequency (Supplementary Fig. 10).

In the case of sweeps arising from SGV, we observed that G12 performed equivalently across hard sweeps from a single origin and the three different starting partial frequencies (Supplementary Fig. 11). This is because, with lower $f_{init}$ the majority of simulations result in hard sweeps (Supplementary Fig. 7). Furthermore, in cases where SGV sweeps are soft, the haplotypes bearing the adaptive mutation are expected to be more similar to each other compared to sweeps from recurrent de novo mutations. This is because, in SGV sweeps, the adaptive mutation can be traced to a single origin. Whereas in multiple origin sweeps, the mutation arises independently on multiple distinct genetic backgrounds.

In addition to G12, we compute the SF2 composite-likelihood ratio (CLR) statistic across hard sweeps, sweeps from recurrent de novo mutations, and sweeps from SGV. The SF2 CLR statistic evaluates the likelihood of an SFS being generated by a hard sweep model (numerator) relative to a neutral model (denominator). To run SF2, we computed the neutral SFS from 500 neutral simulations under the Tennessen et al. model[48]. We then computed the CLR statistic across 1000 bp intervals across all simulated chromosomes for all hard and soft sweep simulations. Finally, we took the highest CLR value from a 10Kb window centered around the selected site of the chromosome. We did this for all 500 hard and soft sweep simulations and estimated the true positive rate (TPR) and 95% confidence intervals at a 1% false discovery rate (FDR) (Fig. 2 and Supplementary Fig. 11). This FDR threshold was in line with our application of G12 as well as the application in ref. 2.

### Running selection scans on modern data mimicking aDNA

We ran G12 on 91 GBR individuals from the 1000 Genomes Project[50] with diploid genotypes called using the standard process and G12 with the same individuals processed using our ancient DNA mimicking approach. We then compared the G12 values at each SNP in both scenarios and calculated the Pearson correlation coefficient between the values and found that they are strongly positively correlated with each other with a correlation coefficient of 0.95 (Supplementary Fig. 12), suggesting that the computed G12 statistic is highly similar for both multi-locus genotypes and pseudo-haplotypes.

In addition to G12 on the mimicked data described above, we also ran SF2, following the methods described in ref. 2. Specifically, we ran SF2 on a 1000 bp window across the chromosomes and recorded the results for all three populations (CEU, YRI, and STU). To find outlier genes that may be representative of putative signals of selection we followed the outlier gene detection pipeline described in Souilmi et al. [2]. We first transformed the raw CLR score to $log10$ transformed CLR scores, then mapped these transformed CLR to protein-coding genes using ENSEMBL[72] database (genome reference version GRCh37) and removed any genes that were not annotated in the NCBI database (ftp://ftp.ncbi.nih.gov/gene/DATA/GENE_INFO/Mammalia/Homo_sapiens.gene_info.gz). We then mapped the scores to genes in a 50 kb window around the gene boundaries. If multiple SNPs were mapped onto a single gene, we used the one with the highest $log10$ transformed CLR score to represent the score for that gene. On these scores, we performed gene length correction and using those gene length corrected scores, which were approximately standard Gaussian, we computed Z-scores. We calculated $p$ values using standard Gaussian quantiles and observed all three populations had a U-shaped $p$ value distribution, in line with some of the results from ref. 2. To perform $q$ value correction[73], we first converted $p$ values to two-tailed $p$ values. To identify outlier genes, we used genes with $q$ value <0.01 (corresponds to FDR of 1%) and Z-score ≥0. Outlier genes for all three epochs are reported in Supplementary Data 1. Using these outlier genes, we identified the CLR values corresponding to a $q$ value of 0.01, which provided genome-wide significance CLR thresholds for all populations.

### Positive control variants used to validate the performance on real data

After examining the application of G12 on simulated data, we used modern data mimicking aDNA data, as well as samples from the H period (which are just 1000 years separated from modern Europeans) to examine whether we were able to identify three major signals of adaptation previously observed in modern Europeans using a variety of different selection scans[45]. These loci have been fine-mapped and extensively characterized functionally in model organisms and are thought to be amongst the strongest signals of selection observed in modern Europeans. We list these genes and the causal alleles in Supplementary Table 2.

### G12 parameter choices and peak calling

To calibrate G12 we iterated over several parameter choices to improve performance. The most significant parameters are window and jump/step. Window refers to the analysis window size in terms of SNPs, and jump is the distance between centers of analysis windows. To find the best combination of window and jump, we ran a grid search and varied the window size from 50 to 400 SNPs with an increment of 25 SNPs each iteration and jump/step from 1 to 10 with an increment of two SNPs. We tried to optimize our process on the three signals of well-characterized adaptation in humans from the previous section on the H population which is closest in time to modern samples. Larger window-sizes resulted in a decrease in G12 values, and larger step sizes resulted in a decrease in SNP density, as larger windows diminish the power of the statistic by averaging over regions that come from different linkage blocks. As the jump increases, fewer and fewer SNPs are used in the computation, as illustrated in Supplementary Fig. 14. Overall, we found that a window of 200 SNPs and a jump of one SNP were optimal for our datasets and enabled us to detect the well-characterized selection candidates at the genome-wide significance threshold. We also found that a window size of 200 SNPs was suitable for data from all time periods, as the mean physical distance (bp) in a 200 SNP window G12 window as well as the number of segregating sites, were quite consistent across epochs, Supplementary Table 3.

### Quality control for removing false sweeps

After running the selection scans and computing G12 at each focal SNP, we performed quality control to remove spurious peaks that could have occurred due to artifacts or issues with the data. One reason a certain genomic position might have artificially high G12 values is if the focal SNP and the SNPs within its window range overlap with regions of low recombination rate in the genome. The first step in post-processing/ quality control in our pipeline was to remove all windows with mean per-window recombination rates in the lowest fifth percentile genome-wide. Second, we also removed windows where the mean fraction of missing individuals (i.e., the mean of the fraction of missing individuals per SNP for all the SNPs in that window) was greater than the threshold value genome-wide, which we calculated as $Q3 + 0.75 \times IQR$ (where $Q3$ is third quartile and $IQR$ is interquartile range) of the mean fraction of missing individuals per SNP for all windows in each epoch.

Third, our ascertainment scheme on the aDNA array results in each window having a variable physical distance. While most windows are of similar length, some windows are in sites where the distance between positions is considerably lower or higher than the average. To show that our post-filtered data is largely unaffected by these issues, we regressed G12 values against window size (measured in the physical distance), missingness, and recombination rate after the percentile-based removal process described above. We saw that the overall variability in the data explained by these three variables combined was less than 5%, suggesting that we had effectively removed their association with G12 values (Supplementary Table 4). A final issue could be

that there are windows where neighboring SNP positions are not captured well by the probes in our ascertainment scheme, and missingness rates are clustered even though the overall missingness rate is similar to other windows. To deal with these issues, we also removed windows that were consistently in the top 20 peaks genome-wide across a set of modern (the CEU, YRI, and STU populations) and the four ancient European populations we analyzed. The rationale for this is that it is quite unlikely that we see the same selective sweep across populations of such different ancestry, and across such a broad range of time, and signals of that nature are highly likely to be due to data processing issues.

### The impact of quality control on regions identified by SF2
In our re-examination of the use of SF2 on our aDNA time transect, we applied SF2 using the same parameters, peak calling, and gene annotation process used in ref. 2 (Methods: Running selection scans on modern data mimicking aDNA). Using their process, on initial examination, we found 113 unique regions across the different time periods that were under the 1% FDR threshold (Supplementary Fig. 15a). This was in contrast to G12, which found only a total of 16 unique peaks across all epochs.

To see if we could understand the reasons for this discrepancy, we examined the impact of two quality control measures we used with our G12 process that were not used in the application of SF2 to data in ref. 2: removing regions of (a) low recombination and (b) high missingness (Methods: Quality control for removing false sweeps). This removal of positions where apparent signals of selection could have arisen due to a lack of recombination or due to a lack of data greatly reduced the total number of unique regions that appeared to be under selection using SF2 from 113 to 30 (Supplementary Fig. 15b).

### Peak calling and gene annotation
We called peaks by identifying windows with G12 values based on a false discovery rate (FDR) which we defined as the fifth highest G12 value in a total of 58,350 neutral simulations (10× the number of independent analysis windows). These simulations were processed as described previously, including the pseudo-haplodization scheme and introducing the levels of missing data observed in our samples. We then identified putative sweep candidates in our scan as those windows with a G12 value above our defined threshold.

As the main statistic we use is a multi-locus genotype-based scan, loci thought to be under selection lie in windows around top-scoring SNPs where the score (G12 statistic value) is high compared to the rest of the genome. One issue with directly using the G12 statistic value at each position to identify SNPs that appear to be selected significantly genome-wide is that many signals of selection at the SNP level are correlated due to LD. We wished to avoid identifying multiple high-scoring SNPs that are in linkage, as they might represent the same adaptive event. In order to account for this, we utilized a greedy clumping algorithm that looks for immediate positions upstream and downstream of a target SNP above a given threshold (https://github.com/ngarud/SelectionHapStats) as possible candidates. We assigned peaks to genes by taking the focal SNP in each peak and running Ensembl Variant Effect Predictor (VEP)[63] and annotated all protein-coding genes within 265 kb distance upstream/ downstream of the target SNP and assigned the closest protein-coding gene for target SNP while annotating the G12 peaks. The results of our analysis per epoch are shown in Fig. 4a.

On the 1.2 million positions captured on our array, we also annotated 47,384 as "potentially functional" sites[8] that lie in categories that overlap for certain SNPs. About 1290 SNPs were identified as targets of selection in Europeans by the Composite of Multiple Signals (CMS) test[74]; 21,723 SNPs identified as significant hits by genome-wide association studies, or with known phenotypic effect (GWAS); 1289 SNPs with extremely differentiated frequencies between HapMap populations (HiDiff), 5387 SNPs which tag HLA haplotypes and 13,672 expression quantitative trait loci (eQTLs). We then examined the distribution of G12 statistic value across these categories of positions (Fig. 4b).

### Enrichment analysis
We first calculated the *p* values based on deviation from neutrality based on simulations. Using G12 values based on neutral simulation data, we calculated the mean and standard deviation of G12 for each epoch. Using the mean and standard deviation we calculated z-score for each focal SNP for every epoch and observed they were approximately standard Gaussian. We then used those z-scores to calculate the *p* values using standard Gaussian quantiles. After this we used these *p* values along with SNP positions as input data to the functional mapping and annotation of genome-wide association studies (FUMA) tool to obtain significant gene sets for each epoch. The gene sets were produced by comparing the genes of interest against sets of genes from MsigDB using hypergeometric tests. We performed this analysis for gene sets from the GWAS and GO functional categories using FUMA[64]. The resulting gene sets are shown in Supplementary Fig. 16.

### Reporting summary
Further information on research design is available in the Nature Portfolio Reporting Summary linked to this article.

## Data availability
The ancient genomes analyzed in this study are available through the Allen Ancient DNA Resource (AADR), version 51, accessible at: https://reich.hms.harvard.edu/allen-ancient-dna-resource-aadr-downloadable-genotypes-present-day-and-ancient-dna-data. Modern genomes were sourced from the 1000 Genomes Project, including the CEU, YRI, and STU sub-populations, which can be accessed at: https://www.internationalgenome.org. All ancient DNA-related data generated during this study are provided in the article and its Supplementary Data 1. The modern genome data can be reproduced using the code referenced in the code availability section.

## Code availability
Code used for running G12 selection scans can be found here: https://github.com/ngarud/SelectionHapStats, Code for running simulations and generating modern data can be found here: https://github.com/mariharris/Ancient_DNA_simulations.

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

## Acknowledgements

Computing support for the project was supported on the Director's discretionary fund at the Texas Advanced Computing Cluster. V.M.N. was supported on a grant for human brain evolution by the Allen Discovery Center program, a Paul G. Allen Frontiers Group advised program of the Paul G. Allen Family Foundation as well as a fellowship from the Good Systems Fellowship for Ethical AI at The University of Texas at Austin. N.R.G. was supported in part by the Paul G. Allen Foundation, Research Corporation for Science Advancement, the University of California Hellman Fellowship, an NSF CAREER award (no. 2240098), and a National Institutes of Health award (R35GM151023). M.H. was supported by the Systems and Integrative Biology Training Grant (NIH-NIGMS 5T32GM008185-33) and the Training Grant in Genomic Analysis and Interpretation (NIH T32HG002536).

## Author contributions

D.P., M.H., N.R.G., and V.M.N. wrote the paper and performed analysis.

## Competing interests

The authors declare no competing interests.
