## [Transparent Peer Review file · Nature Communications]

Leveraging ancient DNA to uncover signals of natural selection in Europe lost due to admixture or drift

Corresponding Author: Dr Vagheesh Narasimhan

Version 0:

Reviewer comments:

Reviewer #1

(Remarks to the Author)

This study utilized ancient DNA to investigate how natural selection has influenced recent human genome. The main contribution of this paper is the introduction of a new summary statistic called G12ancient. The authors validated their methods using simulated data and known examples of natural selection in modern European populations. Some of the validated signals' time scales appear to align with biological expectations. Subsequently, the analysis was used to several new instances of natural selection. Additionally, the study revealed that various complex traits might have undergone selection during different time periods. While the results from the data are generally plausible, I have some major concerns regarding the details of their simulation and some statistical analyses (see below)

Major:

The neolithic sample differs significantly from the later samples in their PC space (Fig 1), indicating population turnover. I am uncertain if Tennesen et al.'s 2012 model is suitable for mimicking the study populations. Isn't Tennesen et al., 2012 model just a simplistic Out-of-Africa model that capture population growth? If the paper's purpose does not involve studying the efficacy of G12ancient in identifying when and in which population selection occurs, this might not be of great concern. However, it seems inappropriate to assume that such a simulation would tell us much about the power or sensitivity if population turnover and movement could play a major role here.

Definition of the soft sweep. Regarding the simulation on page 6, I believe that simulating multiple beneficial mutations may not capture the same signal as what people usually consider a "soft sweep." I am curious about how the authors define a soft sweep. This term is traditionally defined in the soft sweep I, II, and III papers by Pennings and Hermisson. Perhaps the term's meaning has expanded to some extent but using it to encompass polygenic adaptation appears excessive. Figure 2 constitutes a crucial result of this paper, and I cannot shake the feeling that what the authors tested is not "hard sweep" versus "soft sweep.", but instead monogenic vs. polygenic selection. If soft vs hard was indeed the objective, proper simulations involving selection on standing variation should be additionally simulated. Alternatively, the authors could reframe their analysis in terms of polygenic selection versus monogenic selection. Furthermore, the selection coefficients (lower end) used in the simulation seem quite large, compared to other studies.

In Figure 5, the FUMA analysis might be more informative if presented as a table with listed p-values. The Method section on enrichment analysis did not explain how the p-value computation takes place. Details such as the handling of multiple testing are crucial in this regard. Overall, I believe that enhancing the statistical reporting (e.g., R2 not "R2 score") and technical details is necessary to make the results replicable.

I am interested in reading more about the potential caveats of the G12ancient approach, which wasn't discussed. It would be beneficial to compare G12ancient with an SFS-based approach to identify common selective targets detected by G12ancient and those captured by other methods. To this end, it is perhaps also important to directly benchmark G12ancient against G12 using simulated data (not just in 1KG data). Such comparisons are typically expected when introducing a new summary statistic. Furthermore, it is important to evaluate the extent of genomic regions with selection signals identified by G12ancient compared to other methods.

The value of G12ancient is significantly influenced by missingness. Therefore, if missingness in the data is region-specific, it

would impact the selection of outliers. This is not ideal.

I wonder how much of the FUMA results could be explained by the few major targets of selection rather than anything polygenic.

Minor:

I would like to caution that the simulation using 1KG data is oversimplified since missingness in the aDNA sample is not random. Yet, the authors processed the 1KG data in a manner that assumes random missingness. Non-random missingness could affect power and sensitivity. While the results generally make sense, non-random missingness, in conjunction with pseudohaploid treatment, would more severely distort the true haplotypes and potentially obscure the signal.

Reviewer #2

(Remarks to the Author)

In this manuscript, Pandey et al. apply the H12 statistic of Garud et al. (2015) to pseudo-haploid data in order to detect selective sweeps from ancient DNA (aDNA) data, for which genotypes and thus haplotypes are difficult to accurately assay. The authors refer to this application of H12 as G12ancient (the G notation in regard to genotype rather than haplotype data). The design decisions for the empirical application were appropriate, such as the choice of individuals being UDG processed to avoid the effects of post-mortem DNA damage resulting in observed C to T and G to A transitions, ensuring that individuals were chosen to have little contamination, and post-scan filtering of problematic sites, such as those with high levels of missing data. The manuscript was also generally well-written, though some areas can use additional clarity.

Overall, however, I felt like the study was oversold within the abstract, and I found it difficult to uncover the technical advancement in methodology and the novelty of application and findings of the manuscript. In particular, there is no technical innovation in the generation of the G12ancient statistic of the authors, which is instead the application of the H12 statistic to a different form of data (strings of pseudo-haploid alleles instead of strings of alleles on a haplotype or strings of genotype values across SNPs). There is also no direct use of the time-series aspect of ancient DNA within the analysis. The application is to publicly available aDNA data that has been explored with selection scans in prior studies, and the findings of the authors are consistent with the main findings of the dozens of selection scans within Europeans using modern DNA, aDNA, or both data types, with the inclusion of a few novel candidate loci, which is common for such scan articles. There is little discussion of the novel candidates, which makes this manuscript feel more like a simulation study to evaluate the effects of the application of the H12 statistic to pseudo-haploid data. The performance of this statistic is also unsurprising, as detection of sweeps using genotype values across loci (G12) is likely just as difficult as sampling random alleles to create pseudo-haploids, as the variation will necessarily be higher than for haplotype data in both cases.

Because of my perceived lack of innovation in terms of methodology and in terms of findings, my enthusiasm for this manuscript is dampened. I provide detailed comments below.

Major comments:

(1) When I accepted to review this manuscript, I was initially excited about the novel statistic promised in the abstract due to the name G12ancient for application to aDNA. However, nothing about the method is related specifically to aDNA, and really G12ancient should be termed G12pseudo to more accurately reflect the application of H12 (G12) to pseudo-haploid data, regardless of whether it derives from ancient samples or not. There is no difference between G12ancient and H12, as the form of the statistic is identical, but the choice of the string construction to obtain string frequencies that are squared and summed in G12ancient is different from H12. There is no correction for DNA damage that is commonplace in aDNA data (I recognize the analyzed empirical samples did not have such post-mortem damage transitions, but typical aDNA samples do).

(2) The novelty of the selection scan is oversold, as it is repeatedly mentioned that most selection scans using aDNA focus on single site statistics (line 45 of Introduction) that do not account for inter-site correlations that haplotype methods do, and then mentioned on line 354 in the Discussion that a single study (Childebayeva et al. 2022) used a haplotype method (XP-EHH) for such problems. However, this ignores Soulimi et al. (2022), which authors cite as reference 2 in this manuscript, who applied SweepFinder to aDNA sampled from different epochs across Europe to examine patterns of selective sweeps through time. Though SweepFinder is not a haplotype approach, it is modeling the correlation between the tested site and the variation at all other sites on the tested chromosome, and thus is in some sense modeling a proxy to linkage disequilibrium and thus haplotype variation. Similarly, the method of Whitehouse and Schrider (2023; Genetics) also employs haplotype frequency information through time, but directly incorporates the temporal information into a single statistic. Therefore, the authors need to better frame the technical advancement of their method and empirical application in light of recent published work.

(3) The authors indicate that they estimate the timing of selection, but they instead provide broad ranges of time based on the ancient samples. There is no estimate in the typical sense. I believe the authors should find other terms here, as estimate is typically taken as some quantitative value obtained from a formal statistical method.

(4) The authors have not directly assessed statistical power to detect selective sweeps with their H12 application (G12ancient), and only show comparisons of box plots of G12ancient values. Because the selection box plots have overlap

with the neutral for key (weak and recent) selection parameters, it is important for the reader to understand the power to detect selection given a false positive rate cutoff. The authors should present their results as power analysis rather than box plots, so that method performance for detecting selection is more easily understood.

(5) The results of Supplemental Figure 5 are strange, as they indicate that adding missing data leads to substantially better detection of sweeps than no missing data. This result suggests to me that the way missing data is handled creates artifacts here, and it seems highly problematic that a sweep statistic would perform substantially better with lower quality data.

(6) The results of Figure 2 as well as description on line 193 suggest that the G12ancient statistic can detect sweeps with the exception of young sweeps with weak selection. However, this is precisely the range that me might expect for most positive selection to have occurred in humans. For example, one of the most prominent signals of positive selection in humans is at LCT, which I believe has estimates of selection coefficients that are no larger than about $s=0.03$, which is only slightly higher than the smallest value ($s=0.025$) considered for selection in Figure 2, for which the only reasonable detection power would be for selection that occurred roughly 1000 generations ago (29,000 years ago) based on the author experiments. I recognize that the higher selection coefficients may be important for other organisms, but for humans, which is the motivation of this study, I find the simulation results unconvincing for all but the strongest selection signals, many of which we have identified using modern data alone. Moreover, because the focus is on humans here, I would argue that the smallest selection coefficient is at best moderate strength (though may be closer to strong selection based on LCT estimates), and that the authors have not evaluated weak selection.

(7) To determine whether selection has been acting, the authors should obtain a genome-wide significant p-value based on simulations. The authors indicate that they chose the fifth highest G12ancient value obtained by simulating neutral data under the Tennesen et al. demographic model. However, this is still an outlier approach. Does this approach allow the authors to reject the null hypothesis of neutrality based on a p-value that has been properly controlled for false discovery? If so, then it would be helpful if the authors expanded upon this, as it was unclear to me. I assume it is not though, as from reading there were 100 simulated replicates run with sequences of length 500,000 bp, and thus the 5th highest G12ancient value would not be significant after a multiple testing correction, as the authors selected 201 SNPs at random within each simulation replicate and thus each simulation replicate corresponds to a single value of G12ancient.

(8) The column labeled "Genes of Interest" in the table of Figure 4c as well as the annotated genes under the peaks of the Manhattan plot of Figure 4a seem misleading. According to Figure 4c, no peak has fewer than five genes, and some peaks have upward of 57 genes. However, it feels as if the authors have cherry picked genes within each of these regions to highlight. I could be wrong, and these genes are indeed those with the highest G12ancient score within each peak, but that is unclear to me if this were the case. Because all genes under a peak have passed the genome-wide cutoff established by the authors, they are all considered candidates and should all be reported.

Minor comments:

(1) Line 51 of the Introduction, the authors state that most haplotype-based methods require phased genomes. However, if they are haplotype-based, then by definition they require phased genomes.

(2) Line 104 of the Results, it would be helpful if the authors explicitly stated how f_4 statistics were used to homogenize the sample sizes. Is it that the authors were essentially choosing populations for which f_4 was not statistically different from zero, which would reflect no admixture history? Also, what are the four populations used within the f_4 tests? I could not find this in the Methods section.

(3) Line 136 of the Results, the authors cite two articles that apply haplotype-based statistics to unphased data, but this is not nearly comprehensive. I would suggest the authors either be more comprehensive here, or indicate that these are example representative articles. As it reads, these are the several methods that have been developed recently, which is not the case.

(4) Line 138 of the Results, the authors indicate that methods using unphased information might be as powerful as those using phased, but this is generally not the trend in prior studies that compare such methods head to head. These studies almost invariably show that unphased methods have lower power than phased methods.

(5) Line 140 of the Results, the authors indicate that the statistics (G12 unphased version of H12 and other unphased statistics) are unable to be applied to aDNA directly because the low coverage means that heterozygotes are called ineffectively. However, just like G12ancient, the input for G12 and H12 are just frequencies of strings, and thus can still be applied. The authors have not evaluated whether application of G12 to genotype calls from low coverage data would actually perform worse than randomly selecting an allele through pseudo-haploidization. I believe this claim is unwarranted here without investigation.

(6) Line 149 of the Results, the authors stated that that modified G12 to work on pseudo-haploidized aDNA data, but there is no modification necessary, as again, G12 and H12 are just combinations of frequencies of strings, where the choice of string definition has changed. The statistic has not changed.

(7) Line 163 of the Results, the authors indicate that they introduce missingness "and" ancient DNA damage at typical rates to evaluate the effectiveness of G12ancient on ancient pseudo-haploid samples. However, aDNA damage was never introduced into this protocol (in fact, the empirical data has no damaging transitions because of the UDG treatment), and so my understanding from reading the Methods is that only missing data that is characteristic of the aDNA samples was

introduced in the modern samples to mimic aDNA.

(8) Soft sweeps were simulated by introducing a beneficial mutation on K (5, 10, 25, or 50) distinct haplotypes at the onset of selection. However, this is not how selection on standing variation acts, as it is expected that the beneficial mutation is likely residing on multiple haplotypes that are more similar than random haplotypes. Moreover, this simulated processes is not how selection on recurrent mutation acts, because all beneficial mutations are introduced at the same time point here. I therefore do not understand what type of realistic process the soft sweep experiments are evaluating.

(9) Line 233 of the Results and Supplemental Figure 8, the authors evaluate a grid of window sizes and jumps (distance between windows) to calculate G12ancient, and ultimately identify a jump of one as optimal. However, if computation is not an issue, then I do not see why a larger jump size than the minimum of one would be optimal.

(10) Lines 254 and 255 of the Results (Figure 3 caption), the authors refer to "the LCT allele" and the "allele frequency". However, what allele is being referred to? LCT is a gene, with multiple alleles. Is the allele a particular variant at the SNP rs4988245 indicated in the text? The authors should clarify here.

(11) In the table of Figure 4c, what does position mean? Is it the start position of the peak, the central position, or something else? Moreover, given the size of the peak, how does the reader determine the start and stop position of the peak (this would be clear if the position meant the start position of the peak, but not if it meant, for example, the position of the central SNP within the peak).

(12) The procedure for creating pseudo-haploid individuals does not involve read sampling, and instead involves one of two alleles sampled with probability of 0.5 (line 417 of the Methods). However, Supplementary Figure 1 indicates a step that involves read sampling, which is not necessarily the same as sampling one of two alleles with probability of 0.5. Did the authors perform read sampling in any of their experiments? If not, then what would be the effect on G12ancient if reads were sampled rather than alleles from a genotype.

(13) To evaluate the correlation of G12ancient with G12, the authors created pseudo-haploid data from the 1000 Genomes Project with missing data properties that mimic aDNA (section on line 441). How were the 708 individuals chosen from the 1000 Genomes data? Were they all European? If not, then how does the introduced population stratification impact such experiments if any.

(14) In the caption of Supplemental Figure 8, it would be helpful if the authors indicated whether the window and jump sizes were in SNPs, bp, kb, or some other units.

(15) The authors performed post-processing/quality control by removing windows with mean per-window recombination rates in the lowest fifth percentile genome-wide. I get the motivation for doing so, but my understanding is that SNP density correlates with recombination rates, and so the authors may be missing out on true signals of positive selection by performing such a filter. Have the authors investigated these regions that they filtered out to evaluate whether they indeed look like artifacts or problematic regions?

Reviewer #3

(Remarks to the Author)

This paper describes a novel approach for detecting evidence of selection from a genome scan of ancient DNA samples. Genome scans and studies of natural selection form a well-trodden path, but I think there is sufficient novelty here for the study to be of wide interest. Broadly, it seems to be well motivated and well-executed. The authors have compiled a logical array of evidence that their method works, both from applying to datasets where evidence for selection has been previously established, as well as from simulations. I have some minor comments:

ll. 66-67. Perhaps worth expanding this point, or pointing to a citation. I am not sure what is meant, otherwise.

124. BP -> BCE? (at least, judging from the figure).

141. Would there be an issue if a read had more than one SNP (somewhat unlikely, I imagine)?

ll. 150-155. I am still a bit uncertain from this description of what constitutes a 'multisite'? The figures (e.g. Figs 3, 4) suggest localised evidence of selection in the genome, so there must be a window width for the region of interest but it is not immediately obvious from the main text what that is. It is only in the Discussion that the figure of 200 arises, and then I used a text search to find it buried in the Methods.

173. Perhaps in the Methods section a little more explanation on how the genotypes in the first generation of the SLiM simulation were generated, and to what extent this might affect the results (i.e. in terms of coalescence/ancestry/equilibrium).

245. It might be helpful, if appropriate, to provide similar evidence to that in Fig 3c (maybe in a table rather than figure) for sites that have been suggested to be under selection (e.g. Fig 4) (at least, where the putative causal variant is known). Presumably it is feasible to try to associate the contribution of individual variants to the score of their test statistic to identify candidate sites. And with many genes there are only a handful of putative causal variants that are implicated.

Reviewer #4

(Remarks to the Author)

The current Pandey et al. manuscript presents the modification and adoption of a multi-locus summary statistic for detecting natural selection on ancient DNA data. The paper is interesting as it provides a different flavour of ancient DNA selection scans, but to me doesn't provide major breakthroughs in our understanding of selection in the human past.

It is noteworthy that the modification and validation of G12ancient is not specifically for ancient (temporally distributed) data, but for low-coverage "pseudohaploid" genome data. The presentation as being an ancient-DNA-aware method is thus slightly misleading to me. Furthermore, imputation of ancient DNA is increasingly alleviating the central low-coverage problem that the paper aims to address.

Another caveat of the method is that it requires populations to work, and is thus not able to use the temporal structure of ancient DNA. This requires the authors to identify specific subgroups of ancient DNA samples, but these will then have a temporal stratification which could potentially impact the statistic.

Overall though, the paper seems technically sound both in its analysis of single-locus sweeps and polygenic selection, of course with the usual caveats associated with such analysis. It will be of interest to people working on selection studies in low-coverage genomes.

Version 1:

Reviewer comments:

Reviewer #2

(Remarks to the Author)

In this revision, the authors have made a great effort to improve their manuscript by clarifying methods and presenting results in a way that is more easily understood (e.g. power curves instead of box plots for simulation results). However, a couple of my major issues from the original submission still stand, including the lack of technical or methodological innovation and the lack of significant novelty in selection scan findings. In addition to these major issues that have persisted, I still believe that the selection simulations inaccurately model the processes that lead to soft sweeps. Though a nice paper overall, as well as a rigorous and generally well-written study, it feels like a simulation study to evaluate how a single method or statistic might perform with data coming from a different distribution than originally intended. I found that the paper was a pleasure to read, but I do not see how it accomplishes more than what is expected of a study published in a field-specific venue targeting a particular audience.

(1) The simulation study and selection scan are evaluating the effect of randomly choosing a single allele at each SNP and merging the string of alleles at consecutive SNPs as a haplotype (or as a multilocus homozygous genotype if making into genotypes) and using this haplotype (or multilocus genotype) as input to the H12 (or G12) statistic. Unsurprisingly, this pseudo-haploid generation strategy works within H12 (or G12), as H12 (or G12) is also known to work for multilocus genotypes in general. Interestingly, the authors show that detecting sweeps using input from multilocus genotypes is a harder problem than when using pseudo-haploid data as input (Supplementary Figure 6), which is a useful result that may have implications for studies for which genotypes can still be accurately called. However, aside from this finding and the rigorous evaluation of the use of pseudo-haploid data as input to H12 (or G12) through simulations, I do not see any other methodological innovations within the manuscript. The manuscript is motivated for evaluating how to use the low coverage data that might be found in ancient DNA studies, for which individuals are sampled across time. Yet the key features of such studies are the temporal and spatial sampling of individuals, and neither of these features are used as input to the statistic. Thus, from a methodological standpoint, there is still no technical innovation.

(2) The results of the selection scan revealed some interesting findings, with some of the 14 genomic regions found as potentially under selection across all time intervals (e.g. HLA, historically thought to be under selection continually through time), while others were isolated to a specific time interval. Such patterns of time-specific selection signals in the recent past in humans are consistent with other selection statistics applied to ancient samples (e.g. Le et al and Soulimi et al). The authors also found 5 genomic regions with novel candidates of selection related to immunity or obesity, both of which have been phenotypes demonstrated to be associated with selection signals in past studies. However, little discussion is provided regarding these novel candidates. In general, I am still concerned with the lack of significant novelty in the selection scan findings, with the general patterns observed in a couple other previous ancient DNA selection scans and a handful of novel candidates that are superficially explored. I would have been less concerned with this issue if the manuscript had introduced some new method or statistic.

(3) The authors follow the simulation approach of Harris et al (2018) to generate soft sweep data in which beneficial mutations are introduced on K randomly chosen haplotypes at the exact same time point. This is, however, neither how selection on standing variation nor how selection on recurrent mutation act. The authors' response is that this is how Harris et al (2018) did it and so they followed this simulation scheme, but the authors should be simulating selective processes as realistically as possible. Aside from wishing to not redo the simulations, I see no reason why the authors would not have simply rerun their simulations using a realistic model of selection.

(4) There are significant typos throughout the revised manuscript. There are many locations (e.g. line 169) that read "Error~Reference source not found" in bold. There is also a slight typo on line 585 in which the name and caption of Supplementary Table 1 is duplicated under Supplementary Figure 5. These are minor concerns, but given how frequent some of the errors occur, I figured I would mention it.

Reviewer #3

(Remarks to the Author)

This paper seems improved over the original, which I reviewed quite favourably last time round. The authors have addressed my concerns (minor point below). In particular, the updated Fig 2 looks convincing.

Note there is still a discrepancy between " dated from 1973 to 353 BP" and dates that appear to be showing >2000BP in Fig1 a.

Reviewer #4

(Remarks to the Author)

The authors provide responses to my original comments on this paper , but I don't find the revised version to provide substantial advances with respect to the limitations identified in the first round of review. I continue to think that the paper is quite sound and should be published with the new clarification about the method not being ancient-DNA-aware, but I think it is a modest advance that will be of interest mainly to some specialists.

Reviewer #5

(Remarks to the Author)

I had a careful read of both the manuscript and the reviewers' comments. In my opinion, the authors provided a meticulous and convincing response to the concerns of the reviewers - I appreciate their careful effort in validating the G12ancient sweep statistic.

I think that the study is novel and exciting enough to justify publication in Nature Communications.
Here are my reasons:

- It provides independent support for the previously observed pattern of strong ancient selective sweeps that are not found in data from modern populations. This suggests that recent admixture events might have obscured strong but locally restricted selective sweeps that have happened in ancestral populations before mixing.
- It's the first application of a haplotype method to aDNA for detecting adaptation, i.e. the first careful assessment using both simulations and functionally validated variants.
- The application of G12 to pseudohaploid data is a simple but clever idea. The authors go a long way to show that it is robust and suitable for ancient DNA with high missing data rates, ascertainment bias, and DNA damage. This is promising for aDNA studies in non-human organisms where accurate imputation and phasing is often not possible. Even for human data, it provides a robust alternative to methods that heavily rely on accurate imputation/phasing.
- The authors show that the new method is substantially better at detecting selective sweeps than a method that was previously applied to aDNA (SweepFinder2), particularly for soft and partial sweeps. This is not surprising since SweepFinder2 is specifically designed for hard, complete sweeps, but suggests that the new method (G12ancient) is a powerful complement to allele frequency based methods.

Here are my views on the authors' response to the major concerns of the previous reviewers:

- The method is validated based on simulations under the Tennesen et al.'s 2012 demographic model, which doesn't include many of the more complex population separation and mixture events inferred from ancient DNA. However, this is still an appropriate approach for testing the performance of the method in light of pseudo-haploidization, missingness, DNA damage, and a population bottleneck. Further, it seems implausible to me that admixture per se can generate a false-positive sweep signal - it rather introduces more haplotypes than generate regions with low haplotype diversity.
- The way that the authors simulate a soft sweep, i.e. by adding beneficial variants onto segregating haplotypes at a single time point, seems valid to me. It might generate sweeps that are 'too soft' than what one would expect from selection on standing variation. However, the more encouraging is the good performance of G12ancient in detecting those simulated sweeps.
- I appreciate the added comparison to the method SweepFinder2, which is the only other method that was previously applied to pseudohaploid data. This provides a valuable addition to the manuscript.
- The authors now carefully investigate the robustness of their method to missingness and recombination rate variation across the genome. This is a crucial addition to the manuscript and makes it a much stronger study - I don't have any concerns about the validity of their results regarding these factors.

I want to provide some minor additional suggestions:

- The gene set enrichment analysis for GWAS annotations is arguably the weakest part of the study. However, to be fair, not a lot of emphasis is put on these results and they are not mentioned in the Abstract or the Discussion. My issue is that very weak signals are integrated across a large number of genes. A significant test suggests that there is some subtle deviation in overall haplotype distribution across the tested genes. However, those weak signals might not have anything to do with positive selection, e.g. background selection would also subtly shift haplotype distribution across many genes. I suggest to either remove these results or to add a caveat in the discussion concerning the difficulty of interpreting these results.

- Line 137: "likely to be present"

- Line 164: Shouldn't this be 500KB, not 50KB? Elsewhere in the manuscript, it is noted that 500KB windows are simulated.

- It should be noted somewhere that SweepFinder2 is designed to detect hard sweeps that are fixed. It is known from previous studies that power is low for partial sweeps and soft sweeps. This explains its performance in Fig. 2, where for most scenarios the beneficial allele does not have enough time to fix in the population, and might also explain why some of the candidate genes are not detected in modern data using this method.

- In Fig. 4b, can you add a reference distribution of G12 for SNPs that were not annotated as being functionally relevant? Otherwise, it is impossible to judge how much these distributions are elevated.

- Please clarify what the window size of "200 SNPs" exactly refers to - is this based on variants on the 1240k array, variants that are polymorphic in the full analyzed dataset, or variants that are polymorphic in just the population that G12 is computed in. I.e. what is counted as a "SNP"?

Christian Huber

Version 2:

Reviewer comments:

Reviewer #5

(Remarks to the Author)

The authors have substantially improved their manuscript, satisfactorily addressing my previous comments. They demonstrate that multi-locus approaches can be utilized on ancient DNA (aDNA) data. Their analysis reveals that signals of selection, which are detectable in ancient individuals, are obscured in modern individuals due to admixture or genetic drift. Furthermore, the authors have now simulated selective sweeps from recurrent de novo mutations as well as from standing genetic variation (SGV) to replicate more realistic scenarios that generate patterns of soft sweeps. This thorough evaluation strengthens their findings, demonstrating the robustness of their approach.

In my opinion, this work represents a significant advance in the field and warrants publication in Nature Communications.

Christian Huber

Reviewers' comments:

We thank all the reviewers for their insightful comments that have significantly improved the manuscript both in terms of strengthening the scientific results and the writing.

One of the concerns raised by several reviewers was that we were presenting our results as though we were developing a new statistic. However, we agree with the reviewer that we are in effect evaluating the use of a previously described and validated statistic, G12 on pseudo-haploid data along with missingness and damage that are typical of aDNA data. Our choice of using a different name G12ancient was to be consistent with the literature where the computation of H12 (Garud et al. *PLoS Genetics*, 2015) on unphased diploid data was renamed G12 (Harris et al., *Genetics*, 2018). However, on the reviewers' concerns we have now reworded our approach through the manuscript, and just state that we are computing G12 on pseudo-haploid data instead of giving it a completely new name.

Reviewer #1 (Remarks to the Author):

This study utilized ancient DNA to investigate how natural selection has influenced the recent human genome. The main contribution of this paper is the introduction of a new summary statistic called G12ancient. The authors validated their methods using simulated data and known examples of natural selection in modern European populations. Some of the validated signals' time scales appear to align with biological expectations. Subsequently, the analysis was used to several new instances of natural selection. Additionally, the study revealed that various complex traits might have undergone selection during different time periods. While the results from the data are generally plausible, I have some major concerns regarding the details of their simulation and some statistical analyses (see below)

Major:

The neolithic sample differs significantly from the later samples in their PC space (Fig 1), indicating population turnover. I am uncertain if Tennesen et al.'s 2012 model is suitable for mimicking the study populations. Isn't Tennesen et al., 2012 model just a simplistic Out-of-Africa model that captures population growth? If the paper's purpose does not involve studying the efficacy of G12ancient in identifying when and in which population selection occurs, this might not be of great concern. However, it seems inappropriate to assume that such a simulation would tell us much about the power or sensitivity if population turnover and movement could play a major role here.

The reviewer is correct that the Tennesen et al model was built to mimic summary statistics (LD, allele frequency spectrum, F_{st} etc.) of real whole genome sequencing data from the 1000 Genomes Project, representing modern European populations and has been used widely in a number of studies examining demographic history and natural selection in human populations (<https://onlinelibrary.wiley.com/doi/full/10.1002/gepi.22264>, <https://journals.plos.org/plosgenetics/article?id=10.1371/journal.pgen.1005928>, [https://www.cell.com/ajhg/fulltext/S0002-9297\(21\)00055-0](https://www.cell.com/ajhg/fulltext/S0002-9297(21)00055-0), <https://genome.cshlp.org/content/26/7/863.full>). This model doesn't capture the fine grained intricacies of population separation and merging within Europe over the past 10,000 years that has recently become appreciable through ancient DNA sequencing over the past 10,000 years. However, while it is possible to infer proportions of mixture reasonably accurately using ancient DNA, other important parameters such as population sizes and split times are not possible to measure robustly in ancient DNA data due to reduced sample sizes, ascertainment and lack of phased data. Thus demographic models like those of the Tennesen et al. model that capture important statistical features of the genomic data have not been built at the fine grained resolution of the past 10,000 years required to simulate these demographic transitions.

Additionally, the original work on G_{12} (Harris et al) had already examined the effect of admixture on the ability to detect selection using this statistic. Here our goal was to establish the performance of G_{12} on a specific type of data processing. We show that genome-wide statistics produced on diploid data from the 1000 genomes project produce G_{12} values that are correlated with G_{12} values produced on the same dataset but include pseudo-haploidization (random allele calling), missingness, and damage were correlated at $r = 0.95$.

Definition of the soft sweep. Regarding the simulation on page 6, I believe that simulating multiple beneficial mutations may not capture the same signal as what people usually consider a "soft sweep." I am curious about how the authors define a soft sweep. This term is traditionally defined in the soft sweep I, II, and III papers by Pennings and Hermisson. Perhaps the term's meaning has expanded to some extent but using it to encompass polygenic adaptation appears excessive. Figure 2 constitutes a crucial result of this paper, and I cannot shake the feeling that what the authors tested is not "hard sweep" versus "soft sweep.", but instead monogenic vs. polygenic selection. If soft vs hard was indeed the objective, proper simulations involving selection on standing variation should be additionally simulated. Alternatively, the authors could reframe their analysis in terms of polygenic selection versus monogenic selection.

We thank the reviewer for highlighting the confusion regarding our definition of soft sweeps. We believe that this confusion comes from the third paragraph in the section "Evaluating G_{12} on simulated data " where we write that to simulate soft sweeps we

“introduced K beneficial mutations at the time of the onset of selection”. We realize that this is in fact inaccurate, in reality we introduce the same selected mutation (same selection strength at the same site) on K distinct haplotypes which we sample at random at the time of the onset of selection.

We made a few changes across the manuscript to clarify the set-up of our soft sweep simulations where instead of writing “we introduced K beneficial mutations at the time of the onset of selection”, we write “we introduced the adaptive mutation to the same locus at the center of 25 distinct haplotypes sampled at random at the time of the onset of selection. In this scenario, typically 3 distinct haplotypes bearing the adaptive mutation increased in frequency, as the majority rarely survived loss.”. Thus what we are testing here is simply monogenic selection but occurring on one vs many haplotypes. This is distinct from polygenic selection where multiple alleles at different positions along the genome are under selection.

Furthermore, the selection coefficients (lower end) used in the simulation seem quite large, compared to other studies.

We thank the reviewer for this comment. In addition to our previous simulations, we simulated lower selection coefficients $s=0.0075$ and $s=0.01$. However we found that for such selection coefficients the power to distinguish sweeps from neutrality is low (see updated **Figure 2**). The reason for this reduced power comes from the window size that we use in our scan. Longer windows are less likely to generate peaks of homozygosity under neutrality (false positives) but as window size increases $G12$ will be biased towards stronger sweeps. The footprint of a hard selective sweep is expected to extend over $s/[\log(Ne*s)\rho]$ base pairs. On average, a 200 SNP window corresponds to $\sim 490,000$ bp. With such a window size and a recombination rate of $\rho=1e-8$, we should expect to be able to detect peaks with selection strength $\sim s=0.04$. If we were to reduce the window size we would be able to detect weaker sweeps at the expense of more false positives. Given the noisy nature of our data, we decided to use a conservative window size in order to avoid detecting false signatures of sweeps.

We added the previous explanation to the fifth paragraph of the Results: *Evaluating G12 on simulated data* section.

In Figure 5, the FUMA analysis might be more informative if presented as a table with listed p-values. The Method section on enrichment analysis did not explain how the p-value computation takes place. Details such as the handling of multiple testing are crucial in this regard. Overall, I believe that enhancing the statistical reporting (e.g., R^2 not “ R^2 score”) and technical details is necessary to make the results replicable.

We now modify Figure 5 to report the p-values for gene sets across each epoch. We have also added the details of how p-values were calculated including False Discovery Rate correction, which is the standard process for Gene Set Enrichment Analysis approaches such as this in the Methods section under: Enrichment Analysis. Additionally we have made necessary changes to enhance the statistical reporting of the text, we have updated the “R2 score” in Supplementary Table 4 to read “R2”.

I am interested in reading more about the potential caveats of the G12ancient approach, which wasn't discussed. It would be beneficial to compare G12ancient with an SFS-based approach to identify common selective targets detected by G12ancient and those captured by other methods. To this end, it is perhaps also important to directly benchmark G12ancient against G12 using simulated data (not just in 1KG data). Such comparisons are typically expected when introducing a new summary statistic. Furthermore, it is important to evaluate the extent of genomic regions with selection signals identified by G12ancient compared to other methods.

We thank the reviewer for the suggestion of comparing the performance of our method with an SFS-based approach. We now assess the ability of SweepFinder2 to detect hard and soft sweeps compared to *G12*. We found that *G12* has a significantly elevated power to detect both hard and soft sweeps compared to SweepFinder2. We have updated Figure 2 to show ROC curves for *G12* and *SF2* varying the strength of selection and time of sample. We also updated the main results and methods section accordingly to include this new result.

Additionally, we updated Supplementary Fig. 6 and instead of showing H12 versus *G12* on pseudo-haploid data we now show the power of *G12* on multilocus genotype data versus *G12* on pseudo haploid data (before introducing missing data). We observe that *G12* calculated on pseudo-haploid data from older and stronger sweeps shows an increase in power compared to *G12* calculated on MLG data. This is likely due to a higher probability of detecting haplotypes at a high frequency when clustering pseudo-haplotypes with exclusively major and minor alleles, as opposed to MLG's with major-homozygous, minor-homozygous and heterozygous genotypes.

The value of G12ancient is significantly influenced by missingness. Therefore, if missingness in the data is region-specific, it would impact the selection of outliers. This is not ideal.

The missiness we encounter comes from some individuals having low or no coverage at a given position. In a given window in which we calculate $G12$, the mean number of individuals with missing data average across all sites was 0.573.

To detect whether this missiness was region specific, we carried out several types of analysis. First, we examined the distribution of mean missiness in windows, and this appeared to be approximately normally distributed (Supplementary Fig. 3, and reproduced here). Second, we carried out a regression to examine whether there was any association between the $G12$ value and the mean missiness across all windows and found that the variance in the $G12$ value explained by missiness was extremely low (<0.03), though it was significant due to the large sample size (~ 1 million positions) used in the association analysis. Third we examined spatial autocorrelation across windows, with different lags (Supplementary Fig. 4). We see that autocorrelation in missiness across neighboring windows was low (<0.2), and largely constant, again suggesting that missiness was random beyond correlation of a single window. We also show that when we incorporate missiness in modern data using an autocorrelation of 0.2 and introducing non-random region specific missiness, $G12$ was still able to identify positive control signals (Supplementary Fig. 5). Finally, to ensure that windows that had the highest $G12$ values were not windows that had high missiness, we also removed windows where the mean fraction of missing individuals (i.e., the mean of the fraction of missing individuals per SNP for all the SNPs in that window) was greater than the $Q3 + 0.75IQR$ (where $Q3$ is third quartile and IQR is inter quartile range), a typical threshold used to remove outliers.

I wonder how much of the FUMA results could be explained by the few major targets of selection rather than anything polygenic.

FUMA is an interface that uses MAGMA (<https://journals.plos.org/ploscompbiol/article?id=10.1371/journal.pcbi.1004219>) a gene set enrichment analysis tool that was specifically designed to accumulate effects of multiple genes across a gene set. If a single locus or gene has a high effect size, it is theoretically possible that it can significantly influence the results of GSEA. However this is far more likely to occur in smaller gene sets as compared with larger gene sets where the high-effect gene's impact gets diluted among many other genes. The original MAGMA paper (<https://journals.plos.org/ploscompbiol/article?id=10.1371/journal.pcbi.1004219>) includes a power analysis of this issue and suggests that high p-value cutoffs (for GWAS significance) will allow for the detection of selection that is not driven by singular genes but by the aggregate of multiple weakly associated loci. In our analysis, we chose a high p-value threshold of 0.05. In running enrichment analysis we examined enrichment for gene sets determined to be associated with diseases or quantitative traits which are highly polygenic. For the significant results we see in the enrichment analysis presented in Figure

5, the average gene set size is over 285, suggesting that single genes are unlikely to be driving the signal.

Minor:

I would like to caution that the simulation using 1KG data is oversimplified since missingness in the aDNA sample is not random. Yet, the authors processed the 1KG data in a manner that assumes random missingness. Non-random missingness could affect power and sensitivity. While the results generally make sense, non-random missingness, in conjunction with pseudo-haploid treatment, would more severely distort the true haplotypes and potentially obscure the signal.

As mentioned to a similar comment by the reviewer above, we tested for non-random missingness in three ways and found that missingness was largely randomly distributed, though there was a low amount of spatial auto-correlation. Importantly, in our analysis we also removed windows with high missingness, ensuring that peaks that had the highest *G12* values were not the windows with the highest missingness. In addition, to address the additional reviewers concern here directly, we modified the processing of missingness in the 1KG data to mimic the maximum spatial auto-correlation in missingness we observed from the real data (autocorrelation between neighboring windows to be 0.2). This would directly simulate the low levels of non-randomness in the data. We kept all of our other processing and calling of peaks the same and found that this did not affect our ability to identify the several well characterized signals of selection observed in modern Europeans. We show the results of this analysis in Supplementary Fig. 3, Supplementary Fig. 4 and Supplementary Fig. 5 and reproduce this here.

Reviewer #2 (Remarks to the Author):

In this manuscript, Pandey et al. apply the H12 statistic of Garud et al. (2015) to pseudo-haploid data in order to detect selective sweeps from ancient DNA (aDNA) data, for which genotypes and thus haplotypes are difficult to accurately assay. The authors refer to this application of H12 as *G12*ancient (the G notation in regard to genotype rather than haplotype data). The design decisions for the empirical application were appropriate, such as the choice of individuals being UDG processed to avoid the effects of post-mortem DNA damage resulting in observed C to T and G to A transitions, ensuring that individuals were chosen to have little contamination, and post-scan filtering of problematic sites, such as those with high levels of missing data. The manuscript was also generally well-written, though some areas can use additional clarity.

Overall, however, I felt like the study was oversold within the abstract, and I found it difficult to uncover the technical advancement in methodology and the novelty of application and findings of the manuscript. In particular, there is no technical innovation in the generation of the G12ancient statistic of the authors, which is instead the application of the H12 statistic to a different form of data (strings of pseudo-haploid alleles instead of strings of alleles on a haplotype or strings of genotype values across SNPs). There is also no direct use of the time-series aspect of ancient DNA within the analysis. The application is to publicly available aDNA data that has been explored with selection scans in prior studies, and the findings of the authors are consistent with the main findings of the dozens of selection scans within Europeans using modern DNA, aDNA, or both data types, with the inclusion of a few novel candidate loci, which is common for such scan articles. There is little discussion of the novel candidates, which makes this manuscript feel more like a simulation study to evaluate the effects of the application of the H12 statistic to pseudo-haploid data. The performance of this statistic is also unsurprising, as detection of sweeps using genotype values across loci (G12) is likely just as difficult as sampling random alleles to create pseudo-haploids, as the variation will necessarily be higher than for haplotype data in both cases.

We agree with the reviewer that we are in effect evaluating the use of *H12* on pseudo-haploid data along with missingness and damage that are typical of aDNA data. Despite being a small modification, this does change the nature of the statistic, as we no longer have phased genotype data. The naming convention of our statistic was to be consistent with the literature. For example, the application of H12 to unphased diploid data was introduced as G12 in Harris et al., *Genetics*, 2018 to indicate that they were strictly not computing haplotype frequencies. In light of the reviewers comments, we have now changed our wording through the manuscript, and just state that we are evaluating the use of G12 on ancient DNA data.

However, we disagree about the impact of this work. Beyond a single study (Childabayeva et al., *MBE*, 2022), where the use of such approaches were not carefully evaluated and imputation based on modern reference panels was used, haplotype based methods have not been applied to aDNA. Our work provides the first assessment using both simulations and functionally validated variants to show that multi-locus genotype based approaches can be applied effectively to ancient DNA. This is a setting where this approach is particularly useful, given the lower sample sizes at early time depths, where knowledge of the admixture history of the populations is limited and where samples themselves are limited because of considerably lower population density before the advent of agriculture and increased difficulty in obtaining data from older samples. Our approach is also likely to be useful in ancient individuals from more temperate climates where again sample sizes are considerably more limited, and data might be only available in sufficient size from one

time period. Finally, in our revised version of our work, we show that there is only partial overlap between signals detectable in recent time periods and early time periods, suggesting that admixture or drift might have obscured selective signals seen in the past.

Because of my perceived lack of innovation in terms of methodology and in terms of findings, my enthusiasm for this manuscript is dampened. I provide detailed comments below.

Major comments:

(1) When I accepted to review this manuscript, I was initially excited about the novel statistic promised in the abstract due to the name *G12ancient* for application to aDNA. However, nothing about the method is related specifically to aDNA, and really *G12ancient* should be termed *G12pseudo* to more accurately reflect the application of *H12* (*G12*) to pseudo-haploid data, regardless of whether it derives from ancient samples or not. There is no difference between *G12ancient* and *H12*, as the form of the statistic is identical, but the choice of the string construction to obtain string frequencies that are squared and summed in *G12ancient* is different from *H12*. There is no correction for DNA damage that is commonplace in aDNA data (I recognize the analyzed empirical samples did not have such post-mortem damage transitions, but typical aDNA samples do).

Our main purpose in giving our statistic a new name was to differentiate between *H12* which is applied to diploid phased genotypes, *G12* applied to diploid unphased genotypes and our current approach where we are computing the same formulation, but on pseudo-haploid data. While the calculation may be the same mathematically, there is a difference between a multi-locus genotype and a multi-locus pseudo-haploid genotype. This was also the rationale behind the new nomenclature for *G12* instead of *H12*, which we are extending to this new type of data here.

In light of the reviewers comments, however, we have reworded our statements and instead of naming this statistic *G12ancient* we state that we apply *G12* to pseudo-haploid data, modifying the definition of the multi-locus genotype but the calculation used is the same. Additionally, we agree that we do not specifically consider aDNA damage as part of the selection statistic but we evaluate the performance of our statistic inducing damage and missigness at the appropriate levels typical of ancient DNA, as well as its use on ascertained positions on the 1.2M SNP capture array.

(2) The novelty of the selection scan is oversold, as it is repeatedly mentioned that most selection scans using aDNA focus on single site statistics (line 45 of Introduction) that do not account for inter-site correlations that haplotype methods do, and then mentioned on line 354 in the Discussion that a single study (Childebayeva et al. 2022) used a haplotype

method (XP-EHH) for such problems> However, this ignores Soulimi et al. (2022), which authors cite as reference 2 in this manuscript, who applied SweepFinder to aDNA sampled from different epochs across Europe to examine patterns of selective sweeps through time. Though SweepFinder is not a haplotype approach, it is modeling the correlation between the tested site and the variation at all other sites on the tested chromosome, and thus is in some sense modeling a proxy to linkage disequilibrium and thus haplotype variation. Similarly, the method of Whitehouse and Schrider (2023; Genetics) also employs haplotype frequency information through time, but directly incorporates the temporal information into a single statistic. Therefore, the authors need to better frame the technical advancement of their method and empirical application in light of recent published work.

The method of Whitehouse and Schrider (timesweeper) was released in July 2023, after the reviews were returned to us, and initially posted to bioRxiv on April 12th 2023, around the same time our manuscript was initially submitted to the journal. It also involves simulation under a demographic model in order to infer selection. Such demographic models have not been established for ancient DNA due to the sparsity of the data, DNA damage and ascertainment on a SNP capture array. While stating it could be used for such data, Whitehouse and Schrider do not themselves provide any examples or attempt to validate their approach on an ancient DNA time transect and instead use a modern *Drosophila* dataset as their primary application.

We agree that SweepFinder2 is not technically a haplotype based approach in the traditional sense but rather one that is based on the local site frequency spectrum at different distances from a focal SNP. On the reviewers suggestion we compared the performance of SweepFinder2 directly on simulated data, followed by its application on modern data mimicking aDNA data. Across almost all parameters we tested - both on the strength of selection as well as its timing, the performance of sweepFinder was significantly below the performance of *G12*. We also closely followed methods described in Souilmi et al., and show that the positive controls used in our study do not come up as outlier genes in CEU population data (processed to mimic aDNA).

We believe that this direct contrast with the only other applicable method to our data provides a clear context for the advancement of our method.

(3) The authors indicate that they estimate the timing of selection, but they instead provide broad ranges of time based on the ancient samples. There is not an estimate in the typical sense. I believe the authors should find other terms here, as estimate is typically taken as some quantitative value obtained from a formal statistical method.

We agree that we are not estimating the timing of selection here as a parameter but rather are identifying time periods in history where our method has power to detect selection. As such, we have now reworded the portions of the manuscript where we refer to obtaining an estimate.

(4) The authors have not directly assessed statistical power to detect selective sweeps with their H12 application (G_{12}^{ancient}), and only show comparisons of box plots of G_{12}^{ancient} values. Because the selection box plots have overlap with the neutral for key (weak and recent) selection parameters, it is important for the reader to understand the power to detect selection given a false positive rate cutoff. The authors should present their results as power analysis rather than box plots, so that method performance for detecting selection is more easily understood.

We thank the reviewer for this suggestion. We have updated Figure 2 to show the power of G_{12} on simulated aDNA data as a function of selection strength. We also ran SweepFinder2 (SF2) and found that G_{12} has more power to detect both hard and soft sweeps than SF2. These results are now shown in Figure 2. Additionally, supplementary figures 6 and 7 have also been updated to a power analysis of the G_{12} statistic under pseudo-haploidization and missing data.

(5) The results of Supplemental Figure 5 are strange, as they indicate that adding missing data leads to substantially better detection of sweeps than no missing data. This result suggests to me that the way missing data is handled creates artifacts here, and it seems highly problematic that a sweep statistic would perform substantially better with lower quality data.

We see how Supplementary Fig. 5 (now Supplementary Fig. 7) can be confusing and we thank the reviewer for pointing this out. The reason why we expect G_{12} to be inflated when the data has high levels of missing data comes from the original haplotype clustering scheme used to compute the statistic. Originally, a missing allele in a haplotype within an analysis window would be ignored and the haplotype would be clustered with the another haplotype matching at all other non-missing sites. With high missing data this would lead to haplotypes with high missingness to be clustered together as a single haplotype, creating a haplotype at high frequency in the population and consequently inflating G_{12} . In order to mitigate false positive driven by the missingness in our data we took two steps:

(1) We adjusted the clustering scheme such that if a haplotype within the window has more than $m\%$ of missing data, then we do not cluster it with an existing haplotype group. In our work we set this threshold to 90%, that is if a haplotype has more than 90% missing data then we do not include it in a haplotype cluster. We now include the power of G_{12} before

and after this modification in Supplementary Fig. 7 and also include an explanation of this approach in the *Methods: Running selection scans on simulated data* section.

(2) We define the cut-off threshold to call sweeps based on neutral simulations to which we introduce the same levels of missingness as those observed in the data. This threshold is based on a false discovery rate (FDR) which we defined as the 5th highest *G12* value in a total of 58,350 neutral simulations (10X the number of independent analysis windows). Missing data will also increase the baseline *G12* value expected in neutrality which will result in a higher cutoff value than if computed from neutral simulations with no missing data, potentially reducing the false positive rate.

Importantly, we have revised both our simulations and the application on real data to utilize this new approach. In Supplementary Figure 7, we show that the power of our method does decrease significantly with missing data in line with expectation.

(6) The results of Figure 2 as well as description on line 193 suggest that the *G12*_{ancient} statistic can detect sweeps with the exception of young sweeps with weak selection. However, this is precisely the range that we might expect for most positive selection to have occurred in humans. For example, one of the most prominent signals of positive selection in humans is at LCT, which I believe has estimates of selection coefficients that are no larger than about $s=0.03$, which is only slightly higher than the smallest value ($s=0.025$) considered for selection in Figure 2, for which the only reasonable detection power would be for selection that occurred roughly 1000 generations ago (29,000 years ago) based on the author experiments. I recognize that the higher selection coefficients may be important for other organisms, but for humans, which is the motivation of this study, I find the simulation results unconvincing for all but the strongest selection signals, many of which we have identified using modern data alone. Moreover, because the focus is on humans here, I would argue that the smallest selection coefficient is at best moderate strength (though may be closer to strong selection based on LCT estimates), and that the authors have not evaluated weak selection.

We tested lower selection strengths and found that the power of *G12* is low in this parameter regime. To detect weaker sweeps we could reduce the window size used to compute *G12*, but this would be at the expense of increasing false positives from homozygosity under neutrality due to drift. Please see our response to a similar comment by Reviewer 1 above. We now include a more thorough explanation of this in the *Evaluating G12 on simulated data* section.

More broadly, our goal was to show that our method is capable of detecting selection at this higher selection coefficient regime, and is considerably better powered than an alternate method.

In applying this approach to real data, our goal in applying this multi-locus genotype based approach was to find examples of a few strong signals of selection that could have been obscured either by population turnover.

(7) To determine whether selection has been acting, the authors should obtain a genome-wide significant p-value based on simulations. The authors indicate that they chose the fifth highest G_{12} ancient value obtained by simulating neutral data under the Tennesen et al. demographic model. However, this is still an outlier approach. Does this approach allow the authors to reject the null hypothesis of neutrality based on a p-value that has been properly controlled for false discovery? If so, then it would be helpful if the authors expanded upon this, as it was unclear to me. I assume it is not though, as from reading there were 100 simulated replicates run with sequences of length 500,000 bp, and thus the 5th highest G_{12} ancient value would not be significant after a multiple testing correction, as the authors selected 201 SNPs at random within each simulation replicate and thus each simulation replicate corresponds to a single value of G_{12} ancient.

We agree that our description of our FDR correction threshold could be clearer. We have now clarified our approach in our manuscript. Specifically, to define the FDR we ran a total of 58,350 neutral simulations which correspond to 10X the number of independent analysis windows in the G_{12} scan. These simulations were processed as described in the methods, including the pseudo-haplodizing scheme and introducing the levels of missing data observed in our samples. From these simulations we obtained the 5th highest G_{12} value and used it as our significance threshold to call putative sweeps.

We updated the manuscript to have a clearer explanation of our FDR approach. In the *Results: Time stratified selection in ancient Europe* section we now write:

“We defined a genome-wide threshold for significance as the 5th highest G_{12} value obtained from 58,350 neutral simulations equivalent to 10X the number of independent analysis windows in the G_{12} scan”

We also updated the Methods section (G_{12} parameter choices and peak calling) to describe our approach more thoroughly.

(8) The column labeled "Genes of Interest " in the table of Figure 4c as well as the annotated genes under the peaks of the Manhattan plot of Figure 4a seem misleading. According to Figure 4c, no peak has fewer than five genes, and some peaks have upward of 57 genes. However, it feels as if the authors have cherry picked genes within each of these regions to highlight. I could be wrong, and these genes are indeed those with the highest G_{12} ancient score within each peak, but that is unclear to me if this were the case. Because all genes under a peak have passed the genome-wide cutoff established by the authors, they are all considered candidates and should all be reported.

We no longer list individual genes of interest and have modified Fig 4c, to just list the total number of genes within a peak. In Supplementary Data we include a list of all genes associated with each peak. In Figure 4a (the Manhattan plot), we now only annotate genes that have already been previously identified to be under selection such as the LCT allele and the HLA region.

Minor comments:

(1) Line 51 of the Introduction, the authors state that most haplotype-based methods require phased genomes. However, if they are haplotype-based, then by definition they require phased genomes.

We have changed this sentence to read “most haplotype-based methods rely on phased genomes”. Our purpose here was to contrast haplotype based methods with the multi-locus genotype method which we used in this manuscript.

(2) Line 104 of the Results, it would be helpful if the authors explicitly stated how f_4 statistics were used to homogenize the sample sizes. Is it that the authors were essentially choosing populations for which f_4 was not statistically different from zero, which would reflect no admixture history? Also, what are the four populations used within the f_4 tests? I could not find this in the Methods section.

We have added details of how f_4 statistics is used in Methods: *aDNA data curation*.

We used an approach from Narasimhan et al., Science, 2019 to remove individuals with types of ancestry found in low fractions in the peripheries of Europe, who do not fall under the typical model that can explain the vast majority of the ancestry of modern Europeans. Namely that their ancestry comes almost entirely from 3 main ancient sources: European Hunter Gatherers, Anatolian Farmers, and Steppe Pastoralists.

The section now reads: “Within each time period in selecting our subset of 177 samples to use for each time period carried out a qpAdm analysis (which utilize different combinations of pairwise f -statistics) to remove individuals with ancestry atypical of that time period in ancient Europe⁷². Specifically, we removed individuals who could not be modeled by a mixture of Anatolian Farmers, European Hunter Gatherers and Steppe Pastoralists who have made the largest genetic contribution to modern Europeans. This ensured that our samples did not contain elevated levels of Iranian farmer, East Siberian Hunter Gatherer, or African ancestry known to admix into some but not all Europeans”.

(3) Line 136 of the Results, the authors cite two articles that apply haplotype-based statistics to unphased data, but this is not nearly comprehensive. I would suggest the authors either be more comprehensive here, or indicate that these are example

representative articles. As it reads, these are the several methods that have been developed recently, which is not the case.

We have removed the concerned text and rewrote the entire section where we introduce *G12*.

(4) Line 138 of the Results, the authors indicate that methods using unphased information might be as powerful as those using phased, but this is generally not the trend in prior studies that compare such methods head to head. These studies almost invariably show that unphased methods have lower power than phased methods.

We have reworded the entire section on *Detection of selection from aDNA with a multi-locus genotype statistic* and removed that particular line.

(5) Line 140 of the Results, the authors indicate that the statistics (G12 unphased version of H12 and other unphased statistics) are unable to be applied to aDNA directly because the low coverage means that heterozygotes are called ineffectively. However, just like G12ancient, the input for G12 and H12 are just frequencies of strings, and thus can still be applied. The authors have not evaluated whether application of G12 to genotype calls from low coverage data would actually perform worse than randomly selecting an allele through pseudo-haploidization. I believe this claim is unwarranted here without investigation.

Here we used the word *ineffectively* to refer to two things. (1) inability to call diploids. About half of the sites for each individual have coverage ≤ 1 , making it impossible to obtain a diploid genotype at those positions. (2) biases that arise due to calling diploids with extremely low coverage data. Even if we had read coverage >1 , we also run into a major issue of reference bias due to alignment to a linear haploid reference sequence originating from a single individual or a mosaic of several individuals. At each site, this haploid sequence only represents a single allele out of the entire genetic variation of the species. Sequencing reads carrying an alternative allele will have mismatches in the alignment to the reference genome and consequently have lower mapping scores than reads carrying the same allele as the reference. This effect increases with genetic distance from the reference genome, which is of particular interest when using ancient DNA where the populations we are using could be quite distant from modern humans. This reference bias influences variant calling by missing alternative alleles or by wrongly calling heterozygous sites as homozygous for the reference allele. These issues have been previously documented and we now cite 4 references that discuss the impact of coverage on genotype calling accuracy as well as bias from low coverage sequencing data. As these effects are known to influence estimates of heterozygosity and allele frequencies, which could directly impact our analysis of regions under selection, and because the vast

majority of ancient DNA is processed using a pseudo-haploid processing scheme in this study we took the conservative approach of using pseudo-haploid data rather than investigate whether genotype calls could also be used instead.

The references we cite are:

1. Bobo D, Lipatov M, Rodriguez-Flores JL, Auton A, Henn BM. False Negatives Are a Significant Feature of Next Generation Sequencing Callsets. *bioRxiv*. 2016; p. 066043.
2. Ros-Freixedes R, Battagin M, Johnsson M, Gorjanc G, Mileham AJ, Rounsley SD, et al. Impact of index hopping and bias towards the reference allele on accuracy of genotype calls from low-coverage sequencing. *Genetics Selection Evolution*. 2018; 50(1). <https://doi.org/10.1186/s12711-018-0436-4>
3. Chen X, Listman JB, Slack FJ, Gelernter J, Zhao H. Biases and Errors on Allele Frequency Estimation and Disease Association Tests of Next-Generation Sequencing of Pooled Samples. *Genetic Epidemiology*. 2012; 36(6):549–560. <https://doi.org/10.1002/gepi.21648> PMID: 22674656
4. Bryc K, Patterson NJ, Reich D. A Novel Approach to Estimating Heterozygosity from Low-Coverage Genome Sequence. *Genetics*. 2013; p. genetics.113.154500. <https://doi.org/10.1534/genetics.113.154500> PMID: 23934885

In Supplementary Fig. 7 we show the correlation in $G12$ applied to pseudo-haploid data from the 1000 genomes versus $G12$ from diploid data from the 1000 Genomes is 0.95. We also show that the peaks we identify based on diploid data and pseudo-haploid data are identical. This analysis suggests that diploid data from low coverage sequencing yields similar results to pseudo-haploidization.

(6) Line 149 of the Results, the authors stated that that modified $G12$ to work on pseudo-haploidized aDNA data, but there is no modification necessary, as again, $G12$ and $H12$ are just combinations of frequencies of strings, where the choice of string definition has changed. The statistic has not changed.

There is actually no difference mathematically between $H12$, $G12$ and $G12_{ancient}$. They are the same calculation, but applied to data that is diploid and phased, diploid and unphased, and pseudo-haploidized. As mentioned also above in a related comment, we chose to call this $G12_{ancient}$ to keep with the previous nomenclature to ensure that it was clear that that the underlying genotypes we used were pseudo-haploid multi-locus genetic data. However, in accordance with the reviewers suggestion, we have changed “ $G12_{ancient}$ to $G12$ ” at all places in the manuscript.

(7) Line 163 of the Results, the authors indicate that they introduce missingness "and" ancient DNA damage at typical rates to evaluate the effectiveness of $G12_{ancient}$ on ancient

pseudo-haploid samples. However, aDNA damage was never introduced into this protocol (in fact, the empirical data has no damaging transitions because of the UDG treatment), and so my understanding from reading the Methods is that only missing data that is characteristic of the aDNA samples was introduced in the modern samples to mimic aDNA.

The reviewer is correct that UDG treatment does eliminate most of the damaging transitions present in a sample, however a low amount of such transitions are still present (<https://www.ncbi.nlm.nih.gov/pmc/articles/PMC2847228/>). To be completely sure that this does not affect our process, we introduced one C>T substitution every 100 positions. This rate is higher than the excess of C>T substitutions we see when comparing the 1000 genomes diploid data with the mean level of C>T substitutions in our ancient DNA data and is conservative. The updated results adding this level of damage to our data is seen in Figure 2 and Supplementary Figure 11, and suggest that introducing this additional amount of damage has no impact on our findings.

(8) Soft sweeps were simulated by introducing a beneficial mutation on K (5, 10, 25, or 50) distinct haplotypes at the onset of selection. However, this is not how selection on standing variation acts, as it is expected that the beneficial mutation is likely residing on multiple haplotypes that are more similar than random haplotypes. Moreover, this simulated process is not how selection on recurrent mutation acts, because all beneficial mutations are introduced at the same time point here. I therefore do not understand what type of realistic process the soft sweep experiments are evaluating.

We thank the reviewers for their comment. We acknowledge that our soft sweep simulation set-up is not the most realistic simulation of such a process. However, to have more efficient forward simulations we followed the same approach as Harris et al. 2018. We aimed to capture the signature of multiple haplotypes rising in frequency simultaneously, as is expected in a soft sweep.

(9) Line 233 of the Results and Supplemental Figure 8, the authors evaluate a grid of window sizes and jumps (distance between windows) to calculate G_{12}^{ancient} , and ultimately identify a jump of one as optimal. However, if computation is not an issue, then I do not see why a larger jump size than the minimum of one would be optimal.

The reviewer is correct. We did this just to be thorough but agree that one is the optimal choice here.

(10) Lines 254 and 255 of the Results (Figure 3 caption), the authors refer to "the LCT allele" and the "allele frequency". However, what allele is being referred to? LCT is a

gene, with multiple alleles. Is the allele a particular variant at the SNP rs4988245 indicated in the text? The authors should clarify here.

We have changed the text in the figure which now reads frequency of rs4988245 allele in LCT gene. We also have Figure 3 and its caption to reflect the same.

(11) In the table of Figure 4c, what does position mean? Is it the start position of the peak, the central position, or something else? Moreover, given the size of the peak, how does the reader determine the start and stop position of the peak (this would be clear if the position meant the start position of the peak, but not if it meant, for example, the position of the central SNP within the peak).

We have modified figure 4c and it no longer shows the table mentioned in the comment..

(12) The procedure for creating pseudo-haploid individuals does not involve read sampling, and instead involves one of two alleles sampled with probability of 0.5 (line 417 of the Methods). However, Supplementary Figure 1 indicates a step that involves read sampling, which is not necessarily the same as sampling one of two alleles with probability of 0.5. Did the authors perform read sampling in any of their experiments? If not, then what would be the effect on G12ancient if reads were sampled rather than alleles from a genotype.

This was an error in writing. Supplementary Figure 1 illustrates the approach we used in which we actually performed read sampling. At each site we picked one read at random and assigned the sample the genotype of that read.

(13) To evaluate the correlation of G12ancient with G12, the authors created pseudo-haploid data from the 1000 Genomes Project with missing data properties that mimic aDNA (section on line 441). How were the 708 individuals chosen from the 1000 Genomes data? Were they all European? If not, then how does the introduced population stratification impact such experiments if any.

The 708 individuals (177 individuals randomly sampled each to match the sample size per epoch in ancient DNA) from the CEU, GBR, YRI, and STU populations of the 1000 Genomes Project. Thus not all samples are European, but are homogenous within each population. Our approach does not directly utilize data from modern Europeans, and the calculations are done exactly the same in each population, so we do not expect any issues with population stratification here.

(14) In the caption of Supplemental Figure 8, it would be helpful if the authors indicated whether the window and jump sizes were in SNPs, bp, kb, or some other units.

We now state that each of these parameters are referred to in terms of SNPs.

(15) The authors performed post-processing/quality control by removing windows with mean per-window recombination rates in the lowest fifth percentile genome-wide. I get the motivation for doing so, but my understanding is that SNP density correlates with recombination rates, and so the authors may be missing out on true signals of positive selection by performing such a filter. Have the authors investigated these regions that they filtered out to evaluate whether they indeed look like artifacts or problematic regions?

First our threshold for removing regions with low recombination was conservative i.e. we only remove windows that lie in lowest 5% of mean recombination rate per SNP per window genomewide. This threshold is unlikely to remove any true signals. We also examined the pre and post processing results for all epochs. We ran peak calling on both results and associated the top peaks to genes using VEP and found out we were not missing any real signals. Which suggests that our QC pipeline is only removing problematic regions and is not contributing to removing true signals of positive selection.

Reviewer #3 (Remarks to the Author):

This paper describes a novel approach for detecting evidence of selection from a genome scan of ancient DNA samples. Genome scans and studies of natural selection form a well-trodden path, but I think there is sufficient novelty here for the study to be of wide interest. Broadly, it seems to be well motivated and well-executed. The authors have compiled a logical array of evidence that their method works, both from applying to datasets where evidence for selection has been previously established, as well as from simulations. I have some minor comments:

II. 66-67. Perhaps worth expanding this point, or pointing to a citation. I am not sure what is meant, otherwise.

We now reference a paper that specifically discusses the issue of reference bias in genotype calls when mapping low coverage data to a haploid reference genome, and in particular how this affects heterozygosity, and other statistics computed from the data for ancient DNA samples.

124. BP -> BCE? (at least, judging from the figure).

We have used BP everywhere in the manuscript.

141. Would there be an issue if a read had more than one SNP (somewhat unlikely, I imagine)?

We took this to mean reads that have two positions that are different from the reference genome. These variants will be at different genomic positions. In our analysis we treat each position independently so we do not think that there will be an issue here.

ll. 150-155. I am still a bit uncertain from this description of what constitutes a 'multisite'? The figures (e.g. Figs 3, 4) suggest localised evidence of selection in the genome, so there must be a window width for the region of interest but it is not immediately obvious from the main text what that is. It is only in the Discussion that the figure of 200 arises, and then I used a text search to find it buried in the Methods.

We now mention our window size of 200 SNPs in the main results section, right before we describe the application of our method on the ancient DNA time transect.

173. Perhaps in the Methods section a little more explanation on how the genotypes in the first generation of the SLiM simulation were generated, and to what extent this might affect the results (i.e. in terms of coalescence/ancestry/equilibrium).

We agree that we had not included this information in our Methods section. All our simulations include an initial neutral burn-in period of $10N_{e_ancestral}$, where $N_{e_ancestral}$ is the ancestral effective population size of the Tennesen et al. model. We have updated the methods to include this explanation. We add the following sentence:

“To achieve an equilibrium level of genetic diversity, we added a neutral burn-in period of $10N_{e_ancestral}$ to our simulations, where $N_{e_ancestral}$ is the ancestral effective population size.”

245. It might be helpful, if appropriate, to provide similar evidence to that in Fig 3c (maybe in a table rather than figure) for sites that have been suggested to be under selection (e.g. Fig 4) (at least, where the putative causal variant is known). Presumably it is feasible to try to associate the contribution of individual variants to the score of their test statistic to identify candidate sites. And with many genes there are only a handful of putative causal variants that are implicated.

We don't explicitly associate genes to peaks unless the peaks are tagged to well characterized and functionally validated signals of selection from Europe. We report in Figure 4c all the unique peaks we observe across all 4 epochs and their corresponding G12 values.

Reviewer #4 (Remarks to the Author):

The current Pandey et al. manuscript presents the modification and adoption of a multi-locus summary statistic for detecting natural selection on ancient DNA data. The paper is interesting as it provides a different flavour of ancient DNA selection scans, but to me doesn't provide major breakthroughs in our understanding of selection in the human past.

It is noteworthy that the modification and validation of G12ancient is not specifically for ancient (temporally distributed) data, but for low-coverage "pseudohaploid" genome data. The presentation as being an ancient-DNA-aware method is thus slightly misleading to me. Furthermore, imputation of ancient DNA is increasingly alleviating the central low-coverage problem that the paper aims to address.

We agree that our approach does not directly account for temporal structure in the aDNA data. However, we extensively validate our approach on pseudo-haploid data that also includes the levels of missingness, damage and ascertainment schemes typical of ancient DNA. We also compare our approach with another method sweepFinder2, that has also been recently used in a time transect of aDNA and show that our approach considerably better both on simulations and the detection of previously discovered targets of selection.

Until now, imputation has largely not been possible with ancient DNA because previous imputation methods do not work for such low coverage data with any form of reasonable accuracy. Since the paper was returned to us from review, a paper suggesting that imputation could be used on ancient DNA data was published (<https://www.nature.com/articles/s41467-023-39202-0>) which directly assessed the phasing quality of ancient DNA using a newly sequenced high coverage trio. However, the mendelian switch accuracies were lower than that of modern samples and up to 12% for low-coverage data such as the ones we used here. In addition, imputation introduces two types of biases on ancient DNA data. (1) accuracy of imputation varies across population differentiation to a reference panel - samples from more recent time periods in Europe from our time transect impute more accurately on average than older time periods (2) Haplotypes under natural selection in modern Europeans would be represented much more frequently than other haplotypes in the imputation panel. When imputing ancient samples,

it is unclear if the imputation process might introduce sweeps onto ancient samples, simply by imputing a recently selected sweep on modern individuals.

To avoid these types of issues, we chose an approach of pseudo-haploidization which minimizes false positives at the expense of being more conservative. This is in-line with another recent paper (Souilmi, Y., Tobler, R., Johar, A. *et al.* Admixture has obscured signals of historical hard sweeps in humans. *Nat Ecol Evol* 6, 2003–2015 (2022). <https://doi.org/10.1038/s41559-022-01914-9>), which uses sweepFinder2 using pseudo-haploid data on a similar dataset.

Another caveat of the method is that it requires populations to work, and is thus not able to use the temporal structure of ancient DNA. This requires the authors to identify specific subgroups of ancient DNA samples, but these will then have a temporal stratification which could potentially impact the statistic.

We agree that our approach requires population level data to work. However, it also does not require data from multiple time periods - and this has its own advantages. For example, our approach could be applied in settings where it is presently difficult to obtain ancient DNA across multiple time periods due to the climatic conditions, but data from a single site with a great micro-environment for DNA preservation was available. A great example of this comes from South Asia, where temperature and soil conditions are not conducive to long term DNA preservation, with 99% of samples older than 2000 years ago from the region failing to yield authentic ancient DNA data. However, there is excellent data from the Swat Valley, a particularly mountainous region (close to the Himalayas) which was occupied for a period of time during the Iron Age and we have been able to obtain hundreds of ancient DNA sequences. Thus while having disadvantages of not being able to work across samples from different time periods we are able to examine selective events from single time periods for which data might be obtainable from the past.

Overall though, the paper seems technically sound both in its analysis of single-locus sweeps and polygenic selection, of course with the usual caveats associated with such analysis. It will be of interest to people working on selection studies in low-coverage genomes.

REVIEWER COMMENTS

Reviewer #2 (Remarks to the Author):

In this revision, the authors have made a great effort to improve their manuscript by clarifying methods and presenting results in a way that is more easily understood (e.g. power curves instead of box plots for simulation results). However, a couple of my major issues from the original submission still stand, including the lack of technical or methodological innovation and the lack of significant novelty in selection scan findings. In addition to these major issues that have persisted, I still believe that the selection simulations inaccurately model the processes that lead to soft sweeps. Though a nice paper overall, as well as a rigorous and generally well-written study, it feels like a simulation study to evaluate how a single method or statistic might perform with data coming from a different distribution than originally intended. I found that the paper was a pleasure to read, but I do not see how it accomplishes more than what is expected of a study published in a field-specific venue targeting a particular audience.

We thank the reviewer for their comments but would like to offer our perspective on why we think our work has broader implications. First, we feel that our main scientific contribution is not in the development of a new statistic, but rather to demonstrate that multi-locus based approaches can be used effectively on aDNA data which comprises unphased pseudo-haploid data, ascertainment and aDNA damage. This is important for several reasons: (1) The vast majority of ancient DNA data available today is of this nature, (2) Imputation and phasing introduce a certain amount of bias, require data of sufficient coverage, and may not be possible for non-human ancient DNA or certain populations where reference panels are not available. Our paper is the first to carefully evaluate and then apply multi-locus approaches to ancient DNA to uncover selective signals in aDNA. As a result, some of the loci we identify in the most recent time period are indeed not novel by design - we used samples from the most recent 3,000 years to show our approach could indeed discover variants that have been functionally validated. Second, we also apply our approach to times from the past and show that signals of selection that are visible in ancient individuals are no longer seen in scans of modern individuals and have become obscured by admixture or drift. This is an important message and suggests that there are opportunities to uncover many new signals of selection that might have occurred in the past when environmental exposures to pathogens and so on could have been quite different from today. This new insight is also complementary to a recent paper showing similar results using single locus approaches.

(1) The simulation study and selection scan are evaluating the effect of randomly choosing a single allele at each SNP and merging the string of alleles at consecutive SNPs as a haplotype (or as a multilocus homozygous genotype if making into genotypes) and using this haplotype (or multilocus genotype) as input to the H12 (or G12) statistic. Unsurprisingly, this pseudo-haploid generation strategy works within H12 (or G12), as H12 (or G12) is also known to work for multilocus genotypes in general. Interestingly, the authors show that detecting sweeps using input from multilocus genotypes is a harder problem than when using pseudo-haploid data as input (Supplementary Figure 6), which is a useful result that may have implications for studies

for which genotypes can still be accurately called. However, aside from this finding and the rigorous evaluation of the use of pseudo-haploid data as input to H12 (or G12) through simulations, I do not see any other methodological innovations within the manuscript. The manuscript is motivated for evaluating how to use the low coverage data that might be found in ancient DNA studies, for which individuals are sampled across time. Yet the key features of such studies are the temporal and spatial sampling of individuals, and neither of these features are used as input to the statistic. Thus, from a methodological standpoint, there is still no technical innovation.

We agree that our approach does not use a time series directly as an input. However, ancient DNA even from a single time point implicitly has information about when selection occurred, because of radio-carbon dates that can date the time period of the sample. While methods that utilize information from multiple time points may be attractive, they are also limited in that they require large sample size data from multiple time points from the same location. The vast majority of aDNA datasets presently available, particularly from regions of the world outside Europe do not have such data.

Our paper shows that a methodologically alternative multi-locus approach can actually be employed on most of the aDNA data currently available. Such approaches have never been evaluated carefully on aDNA and we provide the first demonstration of this in the paper.

(2) The results of the selection scan revealed some interesting findings, with some of the 14 genomic regions found as potentially under selection across all time intervals (e.g. HLA, historically thought to be under selection continually through time), while others were isolated to a specific time interval. Such patterns of time-specific selection signals in the recent past in humans are consistent with other selection statistics applied to ancient samples (e.g. Le et al and Soulimi et al). The authors also found 5 genomic regions with novel candidates of selection related to immunity or obesity, both of which have been phenotypes demonstrated to be associated with selection signals in past studies. However, little discussion is provided regarding these novel candidates. In general, I am still concerned with the lack of significant novelty in the selection scan findings, with the general patterns observed in a couple other previous ancient DNA selection scans and a handful of novel candidates that are superficially explored. I would have been less concerned with this issue if the manuscript had introduced some new method or statistic.

The main scientific message of our paper is to show that selective events that have occurred in the past have been obscured by admixture or drift. This is an important message that goes beyond individual alleles and our work provides complementary evidence for this from a multi-locus statistic. It is also an observation that was not known before the advent of aDNA. We also identified several novel loci in our scans, but did not want to discuss these in significant detail as mapping identified loci to genes without additional functional follow-up.

(3) The authors follow the simulation approach of Harris et al (2018) to generate soft sweep data in which beneficial mutations are introduced on K randomly chosen haplotypes at the exact

same time point. This is, however, neither how selection on standing variation nor how selection on recurrent mutation act. The authors' response is that this is how Harris et al (2018) did it and so they followed this simulation scheme, but the authors should be simulating selective processes as realistically as possible. Aside from wishing to not redo the simulations, I see no reason why the authors would not have simply rerun their simulations using a realistic model of selection.

We thank the reviewer for their comment. We have now simulated sweeps from recurrent *de novo* mutations as well as sweeps from the standing genetic variation (SGV) to mimic more realistic scenarios as follows:

To simulate sweeps from recurrent *de novo* mutations we introduced adaptive mutations at a rate defined by for $\theta_A = 1$ and $\theta_A = 10$, where $\theta_A = 4N_e\mu_A$, where N_e is the effective population size at generation t and μ_A is the mutation rate of the adaptive mutation. To simulate sweeps from the SGV we introduced a neutral mutation to the center of the simulated chromosome and let it rise in frequency until it reached a partial frequency of $f_{init} = 0.001, 0.005$ or 0.01 . Once f_{init} was reached, the mutation became beneficial.

Sweeps arising from *de novo* mutations can result in hard sweeps if the mutational input of the adaptive mutation is low ($\theta_A \ll 1$). Also sweeps arising from SGV can result in hard sweeps if the frequency of the mutation before the onset of selection is low and the adaptive mutation is present on a single haplotype. To gain insight into the likelihood of a hard versus soft sweep in both of these scenarios, we computed the proportion of hard versus soft sweeps across all simulations. For the recurrent *de novo* mutation scenario we computed the number of independent mutational origins of the adaptive mutation at the time of sampling. Two or more independent origins indicate soft sweeps while a single origin indicates a hard sweep. We found that for $\theta_A = 1$, 77-97% of simulations resulted in soft sweeps, while for $\theta_A = 10$ all simulations resulted in soft sweeps. For the SGV scenario we computed the number of distinct haplotypes bearing the adaptive mutations before the onset of selection. We found that when $f_{init} = 0.001$, most sweeps were hard with only 5-29% of simulations with two or more distinct haplotypes bearing the adaptive mutation. This proportion increased to 51-88% when $f_{init} = 0.005$, and to over 92% when $f_{init} = 0.01$. These results are now shown in **Supplementary Figures 6 and 7**.

We tested the performance of G12 and SF2 of our simulations of sweeps from recurrent *de novo* mutations and sweeps from SGV. As before, we found that G12 performs substantially better than SF2 in all cases. Additionally, as expected, we observed a better performance of G12 on hard sweeps than soft sweeps from recurrent *de novo* mutations. The performance of G12 decreased most for softer sweeps ($\theta_A = 10$) and for sweeps that were too young to have multiple haplotypes at appreciable frequencies (Age of sample = 250 and onset of selection = 280; **Supplementary Fig. 10**). Finally, power to detect soft sweeps from SGV is similar to that of hard sweeps. This is because in cases where SGV sweeps are soft, the haplotypes bearing the adaptive mutation are expected to be more similar to each other compared to sweeps from recurrent *de novo* mutations. This is because, in SGV sweeps, the adaptive mutation can be

traced to a single origin. Whereas in multiple origin sweeps, the mutation arises independently on multiple distinct genetic backgrounds.

We have updated the sections **Results: Evaluating G12 on simulated data** and **Methods: Running selection scans on simulated data** to include the results discussed above. In particular, figures **Fig 2**, **Supplementary Fig. 6**, **Supplementary Fig. 7**, **Supplementary Fig. 10**, and **Supplementary Fig. 11** show the results obtained from simulations of sweeps from recurrent *de novo* mutations and from the SGV.

We also attach the figure summarizing the results of our simulations:

Fig. 2: Receiver operating characteristic (ROC) curves of G12 and SF2 in detecting single origin hard sweeps (red) versus sweeps from recurrent *de novo* mutations (blue) in simulated aDNA data. Each panel shows ROC curves for G12 and SF2 for varying strengths of selection (rows) and age of sample (columns). For the simulations considered here, the onset of selection was set to 1000 generations before present. In **Supplementary Fig. 10** both the age of sample and onset of selection are varied, and **Supplementary Fig. 11** shows the ROC curves for sweeps from SGV. We computed G12 and the SF2 CLR scores in a total of 1,500

simulations (500 hard sweeps, 500 soft sweeps, and 500 neutral simulation) for each combination of parameters with mutation rate $\mu = 1.25 \times 10^{-8}/\text{bp}$, chromosome length $L = 5 \times 10^5$ and recombination $r = 1 \times 10^{-8}$ events/bp.

Supplementary Fig. 6. Softness of sweeps arising from recurrent de novo mutations. (A) Number of mutational origins for $q_A = 1$ and $q_A = 10$ for three different times of onset of selection. **(B)** Proportion of simulations with a single origin (red) and with two or more origins (blue). All simulations were sampled 40 generations before present.

Supplementary Fig. 7. Softness of sweeps arising from SGV. (A) Number of distinct haplotypes bearing the adaptive mutation for three different ages of the mutation at the onset of selection. **(B)** Proportion of simulations with one haplotype (red) and with two or more haplotypes (blue) before the onset of selection.

Supplementary Fig. 10: Power of G12 and SF2 in detecting hard (red) versus soft (blue) sweeps arising from recurrent *de novo* mutations in simulated aDNA data. Simulated aDNA included missing SNPs and pseudo-haploidization. We varied the selection strength of the sweeps (s), the onset of selection (rows) and sample generation (columns). We computed G12 and the SF2 CLR scores in a total of 2,000 simulations (500 hard sweeps, 500 sweeps from *de novo* for $q_A=1$, 500 sweeps from *de novo* for $q_A=1$, and 500 neutral simulation) for each combination of parameters with mutation rate $\mu = 1.25 \cdot 10^{-8}$ /bp, chromosome length $L=5 \cdot 10^5$ and recombination $r = 1 \cdot 10^{-8}$ events/bp. Mean power and 95% confidence intervals measured at a 1% FDR are shown for increasing selection strengths.

Supplementary Fig. 11: Receiver operating characteristic (ROC) curves of G12 and SF2 in detecting single origin hard sweeps (red) versus soft (blue) sweeps from SGV in simulated aDNA data. Each panel shows ROC curves for G12 and SF2 for varying strengths of selection (rows) and age of sample (columns). For the simulations considered here, the onset of selection was set to 1000 generations before present. We computed G12 and the SF2 CLR scores in a total of 2,500 simulations (500 hard sweeps, 500 SGV sweeps with $f_{\text{init}}=0.001$, 500 SGV sweeps with $f_{\text{init}}=0.005$, 500 SGV sweeps with $f_{\text{init}}=0.01$, and 500 neutral simulation) for each combination of parameters with mutation rate $\mu = 1.25 \times 10^{-8}/\text{bp}$, chromosome length $L=5 \times 10^5$ and recombination $r = 1 \times 10^{-8}$ events/bp.

(4) There are significant typos throughout the revised manuscript. There are many locations (e.g. line 169) that read "Error~ Reference source not found" in bold. There is also a slight typo on line 585 in which the name and caption of Supplementary Table 1 is duplicated under Supplementary Figure 5. These are minor concerns, but given how frequent some of the errors occur, I figured I would mention it.

These issues somehow crept into the word version of our paper during the upload process, but are not present in the pdf version. We have now corrected these issues and uploaded just a pdf version for this revision.

Reviewer #3 (Remarks to the Author):

This paper seems improved over the original, which I reviewed quite favourably last time round. The authors have addressed my concerns (minor point below). In particular, the updated Fig 2 looks convincing.

We thank the reviewer for their comments on the paper.

Note there is still a discrepancy between " dated from 1973 to 353 BP" and dates that appear to be showing >2000BP in Fig1 a.

Thank you for catching this. The date ranges written in the paper were incorrect due to a file sorting issue and did not reflect the supplementary data file or the image which is based on the supplementary data file. The youngest sample from the historical period is at 1345 BP. We have now corrected the text to reflect this. The figure is now unchanged as the dates in the figure are correct.

Reviewer #4 (Remarks to the Author):

The authors provide responses to my original comments on this paper , but I don't find the revised version to provide substantial advances with respect to the limitations identified in the first round of review. I continue to think that the paper is quite sound and should be published with the new clarification about the method not being ancient-DNA-aware, but I think it is a modest advance that will be of interest mainly to some specialists.

We thank the reviewer for their comments on the paper.

Reviewer #5 (Remarks to the Author):

I had a careful read of both the manuscript and the reviewers' comments. In my opinion, the authors provided a meticulous and convincing response to the concerns of the reviewers - I appreciate their careful effort in validating the G12ancient sweep statistic.

I think that the study is novel and exciting enough to justify publication in Nature Communications.

Here are my reasons:

- It provides independent support for the previously observed pattern of strong ancient selective sweeps that are not found in data from modern populations. This suggests that recent admixture

events might have obscured strong but locally restricted selective sweeps that have happened in ancestral populations before mixing.

- It's the first application of a haplotype method to aDNA for detecting adaptation, i.e. the first careful assessment using both simulations and functionally validated variants.
- The application of G12 to pseudohaploid data is a simple but clever idea. The authors go a long way to show that it is robust and suitable for ancient DNA with high missing data rates, ascertainment bias, and DNA damage. This is promising for aDNA studies in non-human organisms where accurate imputation and phasing is often not possible. Even for human data, it provides a robust alternative to methods that heavily rely on accurate imputation/phasing.
- The authors show that the new method is substantially better at detecting selective sweeps than a method that was previously applied to aDNA (SweepFinder2), particularly for soft and partial sweeps. This is not surprising since SweepFinder2 is specifically designed for hard, complete sweeps, but suggests that the new method (G12ancient) is a powerful complement to allele frequency based methods.

We thank Dr. Huber for his assessment of the impact of our work.

Here are my views on the authors' response to the major concerns of the previous reviewers:

- The method is validated based on simulations under the Tennesen et al.'s 2012 demographic model, which doesn't include many of the more complex population separation and mixture events inferred from ancient DNA. However, this is still an appropriate approach for testing the performance of the method in light of pseudo-haploidization, missingness, DNA damage, and a population bottleneck. Further, it seems implausible to me that admixture per se can generate a false-positive sweep signal - it rather introduces more haplotypes than generate regions with low haplotype diversity.
- The way that the authors simulate a soft sweep, i.e. by adding beneficial variants onto segregating haplotypes at a single time point, seems valid to me. It might generate sweeps that are 'too soft' than what one would expect from selection on standing variation. However, the more encouraging is the good performance of G12ancient in detecting those simulated sweeps.
- I appreciate the added comparison to the method SweepFinder2, which is the only other method that was previously applied to pseudohaploid data. This provides a valuable addition to the manuscript.
- The authors now carefully investigate the robustness of their method to missingness and recombination rate variation across the genome. This is a crucial addition to the manuscript and makes it a much stronger study - I don't have any concerns about the validity of their results regarding these factors.

I want to provide some minor additional suggestions:

- The gene set enrichment analysis for GWAS annotations is arguably the weakest part of the study. However, to be fair, not a lot of emphasis is put on these results and they are not mentioned in the Abstract or the Discussion. My issue is that very weak signals are integrated across a large number of genes. A significant test suggests that there is some subtle deviation in overall haplotype distribution across the tested genes. However, those weak signals might not have anything to do with positive selection, e.g. background selection would also subtly shift haplotype distribution across many genes. I suggest to either remove these results or to add a caveat in the discussion concerning the difficulty of interpreting these results.

We shared the reviewers' concerns and added two caveats to the interpretation of the results. The first is that in our gene set enrichment analysis we are picking the closest gene to the selective window as the gene under selection and this might not be the case. Second, we also state that it is difficult to interpret these results due to issues associated with background selection or other processes that can cause genome-wide shifts in allele frequency. Following the reviewers suggestion we have also now moved these results from a main figure to a supplementary figure.

- Line 137: "likely to be present"

We have now changed this to "are present"

- Line 164: Shouldn't this be 500KB, not 50KB? Elsewhere in the manuscript, it is noted that 500KB windows are simulated.

We simulated the sweeps using a 500KB chromosome, this was a typo. We have now corrected this.

- It should be noted somewhere that SweepFinder2 is designed to detect hard sweeps that are fixed. It is known from previous studies that power is low for partial sweeps and soft sweeps. This explains its performance in Fig. 2, where for most scenarios the beneficial allele does not have enough time to fix in the population, and might also explain why some of the candidate genes are not detected in modern data using this method.

We agree with the reviewer that SweepFinder2 is designed to identify fixed hard sweeps. We have now added a line stating this to the paragraph on the comparison of our method with SF2 right after the discussion of figure 2.

- In Fig. 4b, can you add a reference distribution of G12 for SNPs that were not annotated as being functionally relevant? Otherwise, it is impossible to judge how much these distributions are elevated.

We have now added this reference distribution to the plot, and the background snps show the lowest mean G12 scores, and windows overlapping the HLA locus showing the highest.

- Please clarify what the window size of "200 SNPs" exactly refers to - is this based on variants on the 1240k array, variants that are polymorphic in the full analyzed dataset, or variants that are polymorphic in just the population that G12 is computed in. I.e. What is counted as a "SNP"?

Our SNPs are all variants that are on the 1240k array regardless of whether they were variable within one of our populations we applied our method to. However, the variants chosen to be on the 1240k array were largely common alleles that are not population specific so the number of non-varying positions should be in the small majority.